# The influence of future changes in springtime Arctic ozone on stratospheric and surface climate

Gabriel Chiodo[1,*], Marina Friedel[1,*], Svenja Seeber[1,*], Daniela Domeisen[1,2], Andrea Stenke[1,3,4], Timofei Sukhodolov[5,6], and Franziska Zilker[1,7]

[1]Institute for Atmospheric and Climate Science, ETH Zürich, Zurich, 8092, Switzerland
[2]Institute of Earth Surface Dynamics, University of Lausanne, Lausanne, Switzerland
[3]Institute of Biogeochemistry and Pollutant Dynamics, ETH Zürich, Zürich, Switzerland
[4]Eawag, Swiss Federal Institute of Aquatic Science and Technology, Dübendorf, Switzerland
[5]Physikalisch-Meteorologisches Observatorium Davos and World Radiation Center, Davos, Switzerland
[6]Ozone layer and upper atmosphere research laboratory, St. Petersburg State University, St. Petersburg, Russia
[7]Swiss Federal Institute for Forest, Snow, and Landscape Research (WSL), Birmensdorf, Switzerland
[*]These authors contributed equally to this work.

**Correspondence:** Gabriel Chiodo (gabriel.chiodo@env.ethz.ch); Marina Friedel (marina.friedel@env.ethz.ch); Svenja Seeber (svenja.seeber@env.ethz.ch)

**Abstract.**

Stratospheric ozone is expected to recover by mid-century due to the success of the Montreal Protocol in regulating the emission of ozone-depleting substances (ODSs). In the Arctic, ozone abundances are projected to surpass historical levels due to the combined effect of decreasing ODSs and elevated greenhouse gases (GHGs). While long-term changes in stratospheric ozone have been shown to be a major driver of future surface climate in the Southern Hemisphere during summertime, the dynamical and climatic impacts of elevated ozone levels in the Arctic have not been investigated. In this study, we use two chemistry climate models (SOCOL-MPIOM and CESM-WACCM) to assess the climatic impacts of future changes in Arctic ozone on stratospheric dynamics and surface climate in the Northern Hemisphere (NH) during the 21st century. Under the high-emission scenario (RCP8.5) examined in this work, Arctic ozone returns to pre-industrial levels by the middle of the century. Thereby, the increase in Arctic ozone in this scenario warms the lower Arctic stratosphere, reduces the strength of the polar vortex, advancing its breakdown, and weakens the Brewer-Dobson circulation. The ozone-induced changes in springtime generally oppose the effects of GHGs on the polar vortex. In the troposphere, future changes in Arctic ozone induce a negative phase of the Arctic Oscillation, pushing the jet equatorward over the North Atlantic. These impacts of future ozone changes on NH surface climate are smaller than the effects of GHGs, but they are remarkably robust among the two models employed in this study, cancelling out a portion of the GHG effects (up to 20% over the Arctic). In the stratosphere, Arctic ozone changes cancel out a much larger fraction of the GHG induced signal (up to 50-100%), resulting in no overall change in the projected springtime stratospheric Northern Annular Mode, and a reduction of the GHG-induced delay of vortex breakdown of around 15 days. Taken together, our results indicate that future changes in Arctic ozone actively shape the projected changes in the stratospheric circulation and their coupling to the troposphere, thereby playing an important and previously unrecognized role as a driver of the large-scale atmospheric circulation response to climate change.

# 1 Introduction

Ozone in the stratosphere plays a vital role in the Earth System, by protecting the biosphere from harmful UV radiation. The distribution of ozone in the stratosphere is determined by two large-scale circulation features. Firstly, the Brewer-Dobson circulation (BDC) transports ozone from the tropics, where it is mainly produced, poleward to the winter hemisphere (Butchart, 2014). Secondly, the polar vortex, a cyclonic jet around the winter pole, acts as a mixing barrier and limits, if sufficiently strong, mixing of ozone-rich air from the mid-latitudes with air masses at higher latitudes (Waugh et al., 2017). Conditions within the cold, isolated vortex favour springtime chemical ozone depletion induced by chlorine and bromine species originating from chlorofluorocarbons (CFCs) (via photolysis) that get activated on the surface of polar stratospheric clouds (PSCs), and consequently catalyse the breakdown of ozone into oxygen (Solomon, 1999). These conditions are met almost every year in the Antarctic stratosphere, due to the cold local temperatures, strong vortex, and weak planetary wave activity in the Southern Hemisphere (SH). As a result of anthropogenic activities, CFCs have become the most important ozone depleting substances (ODSs) over the second half of the 20th century, and have led to massive thinning of the global ozone layer.

Thanks to the signing of the Montreal Protocol (MP) in 1987 and its subsequent amendments, ODSs are being phased out. The decline in ODSs is expected to lead to a recovery of the global ozone layer to 1980 levels by the middle of the 21st century (WMO, 2022). Moreover, declining abundances of ODSs, which also act as greenhouse gases (GHGs), are expected to help mitigate climate change - via a reduction in the projected warming (Goyal et al., 2019; Virgin and Smith, 2019; Egorova et al., 2023; Zilker et al., 2023). Most importantly, the resulting changes in the ozone layer have crucial implications for the Earth system, e.g. reduced exposure of the biosphere to UV radiation (Bais et al., 2018), and protection of the terrestrial carbon sink (Young et al., 2021). A significant increase in ozone levels since the beginning of the century is already detectable in some regions of the stratosphere, such as the Antarctic (Solomon et al., 2016) and the global upper stratosphere (Godin-Beekmann et al., 2022), demonstrating the success of the MP in allowing the global ozone layer to recover.

The recovery of the ozone layer will, however, not occur uniformly across all regions of the stratosphere. In the Antarctic region, the largest increase relative to present-day is expected (WMO, 2022). Outside of this region, ozone will increase in the upper stratosphere and in the Arctic stratosphere, while it will decrease in the lower tropical stratosphere (Keeble et al., 2021). These changes are due to the combined effect of ODSs and GHGs, and the resulting changes in local stratospheric temperature and transport (Chipperfield et al., 2017). Such changes in ozone can also in turn affect the thermal structure of the stratosphere (e.g. via changes in heating), thereby affecting future temperature trends (Maycock, 2016) and inducing a radiative forcing on climate (Bekki et al., 2013). One of the best understood pathways whereby ozone can also affect tropospheric climate is via changes in the meridional temperature gradient in the stratosphere, which alters the dissipation of waves, ultimately resulting in changes of the large-scale atmospheric circulation (Previdi and Polvani, 2014), and regional climate (e.g., Ivanciu et al., 2022), although the details of the physical mechanism are still a subject of ongoing research (WMO, 2022). A number of studies have documented the downward influence of ozone recovery on the region with the largest future changes in ozone: the Antarctic stratosphere (e.g., Ivanciu et al., 2022; Mindlin et al., 2021; Barnes et al., 2014; Polvani et al., 2011; Previdi and Polvani, 2014; Banerjee et al., 2020). These studies consistently showed that the imprint of ozone recovery on the circulation is opposite to

the effects of ozone depletion. Over the recent past, ozone depletion has led to a delay in the breakdown of the stratospheric polar vortex, and a speed-up of the Brewer Dobson circulation (BDC) (Abalos et al., 2019; Polvani et al., 2019). In the future, Antarctic ozone recovery due to declining ODS abundances will slow down the BDC (Polvani et al., 2019) and anticipate the breakdown of the stratospheric polar vortex (Mindlin et al., 2021). These dynamical changes induced by Antarctic ozone recovery extend to the troposphere, resulting in a shift towards the negative phase of the Southern Annular Mode (SAM), an equatorward shift of the mid-latitude eddy-driven jet (Son et al., 2008, 2009; Polvani et al., 2011; Previdi and Polvani, 2014), and changes in rainfall patterns (Purich and Son, 2012; IPCC, 2021).

In the Arctic, the climatological stratospheric air temperatures are warmer than in the Antarctic due to stronger wave activity. These warmer temperatures have limited ozone depletion in the recent past (1980-present). Despite the smaller chemical depletion, strong variations in tropospheric wave forcing also cause sizable interannual variability of ozone levels (Newman et al., 2001). Such large variability in the Arctic stratosphere has also generally made it extremely challenging to detect any significant long-term trends in ozone (e.g., Fusco and Salby, 1999) and, consequently, any dynamical impacts of long-term ozone depletion trends have been considered negligible so far (WMO, 2018, 2022). On shorter timescales, it has recently been demonstrated that Arctic ozone and its inter-annual variations can actively influence stratosphere-troposphere coupling in the context of vortex extremes (Haase and Matthes, 2019; Oehrlein et al., 2020). For example, positive ozone anomalies during sudden breakdowns of the polar vortex (Sudden Stratospheric Warmings, SSWs (Baldwin et al., 2021)) are associated with a negative phase of the North Atlantic Oscillation (NAO) (Domeisen and Butler, 2020). Conversely, episodic depletion events in the Arctic are associated with a strong vortex, and a positive phase of the Arctic Oscillation (Ivy et al., 2017). Extreme variations in Arctic ozone can exert a radiative and dynamical feedback, actively modulating the extremes and dynamical coupling to the troposphere (Friedel et al., 2022a, b).

Over the 21$^{st}$ century, stratospheric cooling from higher atmospheric GHG abundances (primarily $CO_2$), paired with the projected speed-up of the BDC, is expected to substantially change Arctic ozone abundances, leading to ozone levels that in springtime can even surpass historic levels in some GHG scenarios (i.e. "super recovery" (WMO, 2022)). Hence, unlike historic ozone depletion, future trends in springtime Arctic ozone are expected to be significant, particularly in high-emission scenarios (Keeble et al., 2021). Long-term changes in mean ozone levels can affect stratospheric temperature even in the Arctic stratosphere, and can lead to dynamical changes in the stratosphere that project onto surface climate; however, this has only been shown in the context of abrupt-4×CO2 experiments (Chiodo and Polvani, 2019; Li and Newman, 2022), with little applicability for more policy-relevant scenarios from the Intergovernmental Panel on Climate Change (IPCC). In this work, we seek to better understand the role of future changes in springtime Arctic ozone in climate over the 21$^{st}$ century in the Northern Hemisphere (NH), isolating these changes from the effects of GHGs. Our paper aims at documenting the dynamical impacts of projected long-term ozone changes in the Arctic and NH stratosphere and so, by definition, it contains the contribution of all major GHG and ODS emissions to the ozone changes, irrespective of the individual emissions from each of these separate sources.

To our knowledge, there is no study investigating the impact of future changes in Arctic ozone (including the effects of ODS-driven recovery and GHG-driven super-recovery) on the large-scale circulation and surface climate in the NH. Currently,

there is also no consensus about the overall effect of climate change on the stratospheric polar vortex (Manzini et al., 2014; Karpechko et al., 2022), which has implications for regional climate over the NH (Simpson et al., 2018; Ayarzagüena et al., 2020; Karpechko and Manzini, 2012). At the same time, the demand for reliable climate projections at the regional scale is rising. However, compared to thermodynamic aspects, the dynamic response of the climate system to rising GHG levels is only poorly understood (Shepherd, 2014). A detailed assessment of the role of future drivers in atmospheric circulation changes, such as ozone and GHGs, might help to reduce the considerable uncertainty associated with regional climate change projections. In this paper we present modeling evidence pointing at a sizable role of future changes in springtime Arctic ozone in stratospheric and surface climate.

## 2  Methods

In the following, we describe the numerical model simulations and statistical analysis employed in this work.

### 2.1  Model simulations

We use two chemistry-climate models (CCMs); the Whole Atmosphere Community Climate Model (WACCM) and the SOlar Climate Ozone Links (SOCOL). WACCM is the atmospheric component of the NCAR Community Earth System Model version 1 (CESM1.2.2), it has a high top (140 km) and vertical resolution of 66 levels (Marsh et al., 2013) and is coupled to interactive ocean and sea ice components. WACCM has a horizontal resolution of 1.9° in latitude and 2.5° in longitude (Marsh et al., 2013) and can be run in different configurations for ozone, namely with the standard "interactive" configuration (where ozone chemistry is interactive and thus the modelled ozone responds to external forcings and the circulation) and a "specified chemistry" configuration (Smith et al., 2014), in which ozone concentrations and other radiative species are prescribed in the radiation scheme. This model captures stratospheric trends and polar vortex variability well and has been used in a number of studies on the effects of ozone on stratospheric and tropospheric climate (to name a few: (Haase and Matthes, 2019; Oehrlein et al., 2020; Rieder et al., 2019; Friedel et al., 2022a, b)).

The second model we use is SOCOL: this is a CCM (Stenke et al., 2013) based on the general circulation model MA-ECHAM5, which is interactively coupled to the chemistry transport model MEZON (Model for Evaluation of oZONe trends; (Egorova et al., 2003)) and to the ocean–sea ice model MPIOM (Muthers et al., 2014). SOCOL-MPIOM has a model top at 0.01 hPa and 39 vertical levels and is used here at a horizontal resolution of T31 (3.75° x3.75°) (Muthers et al., 2014). SOCOL-MPIOM can be run with interactive chemistry and just like WACCM, SOCOL-MPIOM can also be run with specified ozone concentrations, by decoupling the chemistry module and general circulation model (Muthers et al., 2014). Like WACCM, SOCOL-MPIOM captures stratospheric variability reasonably well (Muthers et al., 2014) and has been recently used for studying the effects of ozone feedbacks on climate (Friedel et al., 2022a, b).

We perform two experiments with 5 realizations each (which solely differ in the initial conditions) for each of the two models, listed in Table 1. First, we run a reference future scenario covering the 21st century (2005-2099) following the high-emission Representative Concentration Pathway 8.5 (RCP8.5) (Meinshausen et al., 2011) for GHGs, while ODSs are following

the recommended WMO 2018 scenario baseline A1 (WMO, 2018). In this scenario (henceforth referred to as "RCP8.5"), the chemistry is fully interactive and thus the changes in the ozone layer from declining ODSs and rising GHGs is simulated, along with its impacts on climate. The second set of experiments (henceforth termed "RCP8.5_fO$_3$") follows the same RCP8.5
scenario for GHGs, but we do not use interactive ozone chemistry. Instead, we impose a monthly-mean 3-D ozone climatology to the models' radiation schemes, perpetually throughout the entire simulated period: this ozone data-set is derived as the ensemble mean from the first 15 years of the RCP8.5 ensemble. In both models, we nudge the QBO by mapping the observed QBO cycles over 1954-2009 into the future. The QBO effects on ozone are thus not fully considered, but since the QBO does not change in any of our future scenarios, this inconsistency is not expected to affect the results.

In both models, the climatology and variability of stratospheric, tropospheric, and surface climate is nearly identical in both configurations in the early stages of the simulations (interactive vs. prescribed, but consistent with boundary conditions) (Smith et al., 2014), even under present-day conditions (Friedel et al., 2022a). We verified this by comparing the two ensembles (the RCP8.5 experiments with interactive ozone vs RCP8.5_fO$_3$ experiments with prescribed ozone) over the reference period used to obtain the ozone climatology imposed in the RCP8.5_fO$_3$ ensemble (2005-2020). This comparison reveals only marginal
differences of less than 1 K in the upper stratosphere (above 10 hPa, not shown), which are likely due to the underestimation of the heating arising from the diurnal ozone cycle (which is not captured by the monthly-mean 3-D ozone climatology, as shown in Smith et al. (2014)). However, these differences are much smaller than the dynamical impacts of long-term ozone trends in the Arctic and global stratosphere (not shown). We also evaluate potential inconsistencies arising from the use of a fixed (constant) 3-D daily ozone climatology (which is derived by averaging over 2005-2020) in an ensemble with transient GHG
forcings (RCP8.5_fO$_3$); see extended discussion in Appendix A. The very small projected changes in the vortex morphology in our two CCMs under RCP8.5 ensures spatial coherence between the vortex and the 3-D ozone climatology even in the late stages of the simulations (2080-2099), thus minimizing any potential artifacts (Fig. A1).

In the high-emission scenario used here (RCP8.5), future ozone changes are considered in one ensemble (RCP8.5) but not in the other (RCP8.5_fO$_3$). Hence, differences between modeled projections by 2100 in the two ensembles (RCP8.5 minus
RCP8.5_fO$_3$) allow us to unambiguously quantify the impact of long-term Arctic ozone changes with respect to present-day, which is the key purpose of this paper.

We carry out five realizations (with fully coupled ocean) for each of the ensembles and models: this allows us to ensure robustness of the results, as discussed in Section 2.2. We note that the future ozone changes considered here include the trends induced by GHGs in a high-emission scenario, aside from the phase-out of ODS. The GHG emissions considered in the
scenario studied in this paper induce a super-recovery in ozone with respect to 1980 levels (WMO, 2022).

## 2.2 Statistical analysis

To assess changes in our simulations over the course of the 21$^{st}$ century, we compare the climatology in the first 20 years of simulations (2005-2024) to the climatology in the last 20 years of the century (i.e. 2080-2099) for the variables of interest. Changes derived that way for RCP8.5 simulations show the combined effect of GHGs and ozone changes on the climate system,
while changes in RCP8.5_fO$_3$ simulations show the isolated effect of GHGs. Thus, differences in changes calculated this way

| Ensemble | Period | Realizations | GHG Forcing | Ozone |
|----------|--------|--------------|-------------|-------|
| RCP8.5 | 2005-2099 | 5 | RCP8.5 | Interactive (transient) |
| RCP8.5_fO$_3$ | 2005-2099 | 5 | RCP8.5 | Climatological (2005-2020) |

**Table 1.** List of model simulations used in this work. Note that both ensembles have been performed with both chemistry climate models, and are fully coupled to the ocean. Note that the climatological ozone forcing is derived from the first 15 years of the RCP8.5 ensemble mean.

between RCP8.5 and RCP8.5_fO$_3$ scenarios display the isolated impact of future ozone changes on the climate system. The analysis presented here is entirely focused on springtime (March – April averages), when Arctic ozone is expected to increase the most due to the projected decline of springtime ozone depletion over the 21$^{st}$ century (Eyring et al., 2013) (their Fig. 6). Therefore, the dynamical and radiative impacts of future ozone changes are expected to maximize in this season, and we thus focus our analysis on springtime. Indeed, this is the season in which signals in the NH are statistically significant (see Fig. B3). We use a two-tailed Student's t-test to assess significance of changes between the first and last 20 years of simulation, assuming independence between two consecutive spring seasons. With n=100 samples (5 ensemble members $\times$ 20 years), the sample size is assumed to be sufficiently large to be distributed normally.

We use the Northern Annular Mode (NAM) at 10 hPa as a measure for large-scale dynamical changes in the stratosphere. For this purpose, empirical orthogonal functions (EOFs) are calculated based on zonally averaged geopotential height anomalies. To ensure that the NAM time series reflects the long-term changes in the mean state of the stratosphere, geopotential height anomalies are calculated for each month of the year as deviations from the monthly climatology from 2005 to 2020. The EOF spatial loading pattern is then calculated based on the same period (2005 – 2020) of geopotential height anomalies north of 20° N, applying latitudinal weights according to the square root of the cosine of latitude. Subsequently, geopotential height anomalies for the whole time period (2005 – 2099) are projected onto the EOF loading pattern to derive the principal component (PC) time series. The PC time series is then scaled to unit variance to obtain NAM indices.

The final stratospheric warming (FSW) is calculated based on springtime wind reversal at 50 hPa and 60° N. More specifically, we define the FSW date as the first day of the year when zonal mean zonal wind at 60° N has fallen below $7\,\mathrm{m\,s^{-1}}$ and does not return above this threshold for more than 10 consecutive days until the following fall. We here adjust the definition proposed by Butler and Domeisen (2021), by using a wind threshold of 7 instead of $5\,\mathrm{m\,s^{-1}}$ to account for biases in our models (Friedel et al., 2022b). We focus on the FSW date in the lower stratosphere (50 hPa), because changes in ozone have been shown to have the largest influence on the FSW in this region (Friedel et al., 2022b).

## 2.3 Radiative impacts of projected future ozone changes

To diagnose the radiative impacts of future ozone changes across our model experiments, we perform offline radiative forcing (RF) calculations with the Parallel Offline Radiative Transfer (PORT) from the Community Earth System Model (CESM) (Conley et al., 2013). First, we carry out a "baseline" PORT run with ensemble mean values for meteorological variables (e.g., temperature, humidity, cloud fraction and height), the spatial distribution of radiatively active species (e.g., $CO_2$, ODS, and ozone), and the zonal mean tropopause height (following the WMO tropopause definition) specified from the average over the period 2005–2010 of the transient RCP8.5 simulations from WACCM and SOCOL. Second, we run a set of "perturbation" runs with PORT, replacing solely the 3-D ozone field, allowing us to obtain the RF and temperature adjustment. The ozone perturbation imposed in PORT is obtained as the projected ensemble mean ozone change over the 21$^{st}$ century, calculated as average differences between the 20-year periods 2080-2099 and 2005-2024, for consistency with the free running experiments. We use hourly instantaneous input meteorological and composition fields (averaged over the five available members), following the approach of Conley et al. (2013) to ensure accuracy in the calculations. All meteorological variables, including water vapor and cloud optical properties, are specified at the year 2005–2010, since changes in such quantities are part of the rapid adjustments and are not part of the stratosphere-adjusted RF.

For each PORT experiment, we compute the annual and global average tropopause-level shortwave and longwave fluxes after stratospheric temperatures reach equilibrium. We diagnose the (radiative) temperature correction needed to achieve radiative equilibrium within the stratosphere, while tropospheric temperatures are kept fixed. This is referred to as the radiative temperature adjustment in the fixed dynamical heating approximation, following the approach by Fels et al. (1980). This approach allows us to quantify the radiative impact on stratospheric temperatures of future ozone changes simulated by the two CCMs, under the assumption that the dynamical heating of the stratosphere does not change. By contrasting this temperature change against the total temperature response to future Arctic ozone changes simulated in the free running experiments in springtime, we can assess the importance of individual processes, namely radiative vs. dynamical heating arising from circulation changes. For the PORT experiments, we consider a longer averaging period (March – May), to take into account the radiative damping time-scale for the lower stratosphere, which is approximately one month and beyond (Ming et al., 2017).

## 3 Results

### 3.1 Climate change under the 21$^{st}$ century and impacts on the atmospheric circulation

In the high-emission scenario RCP8.5, our models project a global warming of 3 K (WACCM4) and 5 K (SOCOL-MPIOM) by the end of the century (Fig. B1 - panel a): these values are within, albeit at the two extremes, of the range of warming projections published in the IPCC-AR5 (Chapter 7 in IPCC (2013); see also Fig. 1 in Sherwood et al. (2020)) and are consistent with the different climate sensitivity of the two underlying models (CCSM4 and MPI-ECHAM5 in Grise and Polvani (2014)). In the latest assessment of the IPCC (AR6), the projected global warming for high-emission scenarios is stronger, but the spread across models is also larger (IPCC (2021), Chapter 4, see Fig. 4.35), due to larger climate sensitivity as well as the stronger

GHG forcing in AR6 projections. Hence, our model simulations are well within the model uncertainty and bracket a good fraction of the inter-model spread, thereby serving as a valid test-bed for assessing the climatic impacts of future Arctic ozone changes.

Aside from surface warming, our models also simulate the well-known pattern of tropospheric warming and stratospheric cooling, resulting from enhanced atmospheric GHG levels (McLandress et al., 2014). In particular, all ubiquitous features of
215 the atmospheric response to $CO_2$ documented across different generations of climate models are simulated by our two CCMs, such as the amplified warming in both the upper tropical troposphere and near the surface in the NH high latitudes (IPCC, 2021). These patterns are very pronounced in NH spring in the RCP8.5 scenario (Fig. 1) but are relatively insensitive of the season, except for the surface amplified polar warming, which maximizes in NH fall (not shown). These patterns also scale quasi-linearly with global warming, with tropical upper tropospheric warming of up to 12 K in SOCOL, being much more
pronounced than in WACCM. In the stratosphere, the cooling maximizes with up to -8 K above 10 hPa (being dominated by $CO_2$) in both models, consistent with previous modeling evidence (Shine et al., 2003). In contrast, polar lower stratospheric temperatures do not change significantly, again consistent with the majority of IPCC-AR6 models (IPCC (2021) - Chapter 4, Fig. 4.22). As a result of these temperature changes, the meridional temperature gradient decreases near the surface, while it increases near the tropopause. By the thermal wind relationship, the subtropical jet strengthens in both hemispheres, with
the largest wind changes again visible in SOCOL (Fig. 1c), due to the stronger tropospheric warming in this model. The strengthening is mostly pronounced on the upper flanks of the subtropical jets, allowing more wave activity to penetrate and dissipate into the subtropical lower stratosphere, ultimately leading to faster tropical upwelling in both models (Fig. B1 - panel b). This mechanism is very robust (Garcia and Randel, 2008; Shepherd and McLandress, 2011) and is reproduced by our two models, and it is relatively insensitive to the season being considered in both models (Fig. B3 - panels c-d).

The subtropical jets extend poleward and upward; under climate change, their upper flank strengthens. These wind anomalies reach the mid stratosphere during NH spring, similar to other models (McLandress et al., 2014). Regarding the NH stratospheric polar vortex, whose strength is usually defined as the zonal mean zonal wind at 10 hPa at 60° N, both CCMs show very distinct responses; SOCOL exhibits a statistically significant strengthening of up to 4 m s$^{-1}$, while no change is simulated in WACCM. Such a lack of a robust response in the polar vortex has been a long-standing issue in inter-model comparisons (Manzini
et al., 2014; Karpechko et al., 2022), even regarding the sign of the projected changes. More generally, the fate of the polar vortex under climate change is uncertain due to competing effects of stratospheric cooling by GHGs, changes in tropospheric wave driving, and stratospheric changes such as the speed-up of the Brewer Dobson circulation. To date, the role of different processes and underlying drivers across models remains an open question (Manzini et al., 2014; Simpson et al., 2018). Future changes in Arctic ozone are yet another potential and undocumented driver modulating the projected polar vortex changes over
the 21$^{st}$ century, which we examine next.

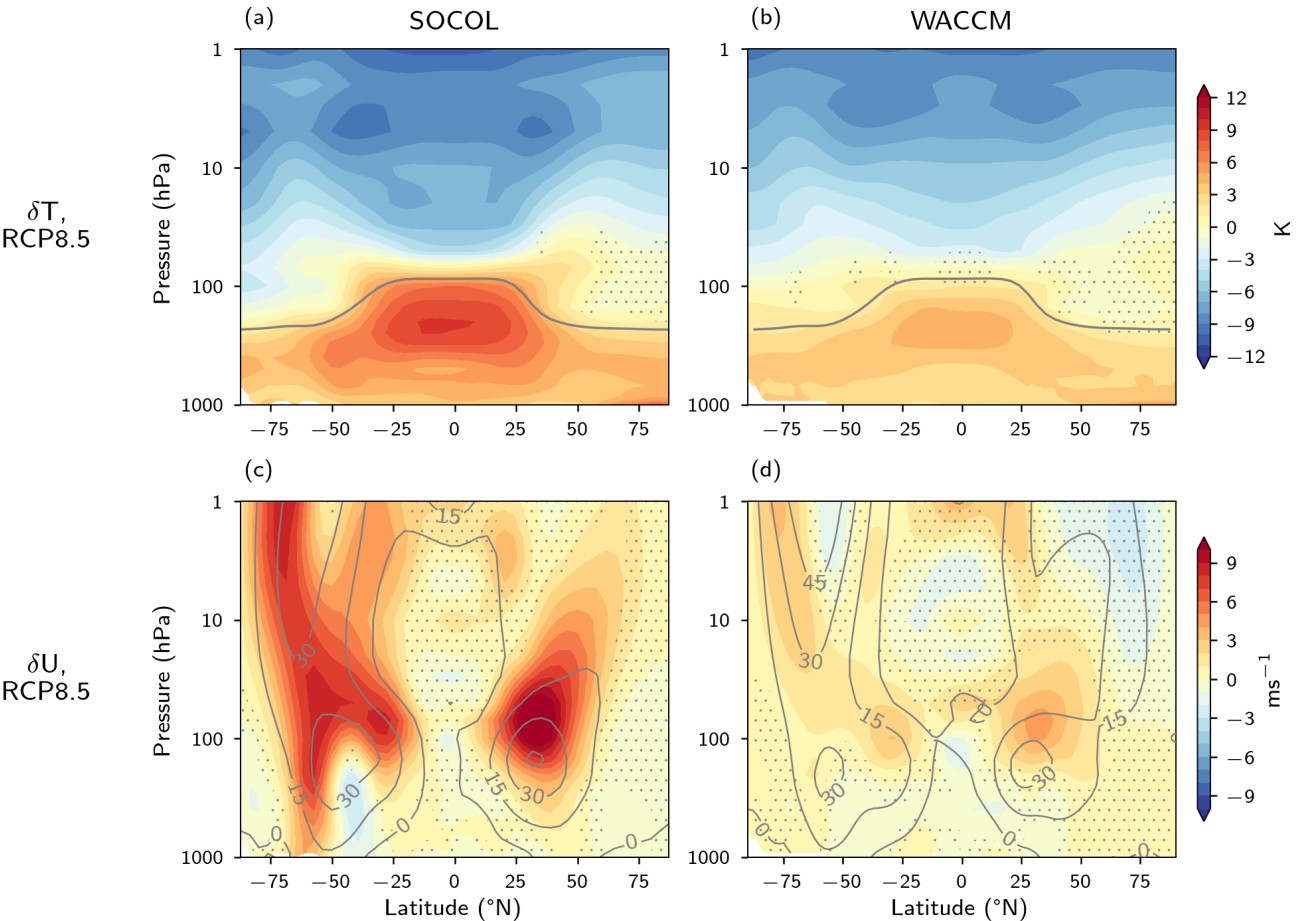

**Figure 1.** Zonal mean temperature (a, b) and zonal wind (b, c) changes in springtime (March-April) between the late (2080-2099) and early (2005-2024) 21$^{st}$ century under RCP8.5 in SOCOL (left) and WACCM (right). Contours in a, b show the tropopause height, and contours in c, d depict the climatology of the zonal mean zonal wind (in m s$^{-1}$) in the early 21$^{st}$ century. Stippling marks regions that are not significant on a 95 % level.

### 3.2 Future ozone changes and impacts on stratospheric climate

Before turning our attention to the impact of future Arctic ozone changes, we first characterize the projected changes in this quantity in the Arctic region and global stratosphere. The simulated zonal mean springtime ozone changes simulated by the two CCMs are shown in Fig. 2. In the upper stratosphere, ozone increases almost uniformly across all latitudes in both CCMs by 20-30 %. Conversely, ozone in the lower stratosphere (50 - 100 hPa) decreases in low latitudes, while it increases in the Arctic. The overall pattern is robust across the two CCMs and is similar to other CMIP6 models in the comparable SSP 5.85 scenario (see Keeble et al. (2021) - their Fig. 10), although sizable differences appear in certain regions, such as e.g. the larger decline in SOCOL in the tropical lower stratosphere and the larger Arctic ozone increase in the mid-stratosphere

in WACCM (i.e. 10 - 30 hPa). The mechanisms behind these projected ozone changes are well understood (WMO, 2022); namely, the radiative cooling in the upper stratosphere from higher $CO_2$ abundances slows down the odd oxygen loss cycles, resulting in less ozone loss and thus an increase in ozone abundances at these stratospheric levels (Haigh and Pyle, 1982; Jonsson et al., 2004). In addition, the decline in ODS leads to less chlorine and bromine-induced ozone depletion in the middle and upper global stratosphere, as well as in high latitudes (where under present-day conditions, the largest ozone depletion occurs). In the high-emission scenario considered here, increases in methane abundances ($CH_4$) also contribute to enhanced Arctic ozone abundances, although increased $N_2O$ acts in the opposite direction (Revell et al., 2012; Butler et al., 2016). In the lower stratosphere, changes in ozone are primarily driven by transport: the speed-up of the BDC (Shepherd and McLandress, 2011) and its upward expansion (Match and Gerber, 2022) lead to enhanced advection of ozone-poor air into the lower tropical stratosphere (Chiodo et al., 2018), further contributing to Arctic polar cap ozone abundances, especially during NH springtime. The larger tropical lower stratospheric ozone decline in SOCOL is likely due to the larger tropical upwelling response, associated with the larger tropospheric warming in this model (Fig. B1). In WACCM, the slightly larger ozone increases in the upper stratosphere at mid-latitudes are possibly leading to slightly larger increase in Arctic ozone abundances than in SOCOL (i.e. a larger "source" of ozone for the dynamical resupply into the Arctic stratosphere is available). These changes in springtime ozone are sizable and can largely affect heating and thus stratospheric temperatures (Maycock, 2016), but such effects cannot be easily quantified in these transient scenarios alone, as they include several forcings and drivers acting in parallel.

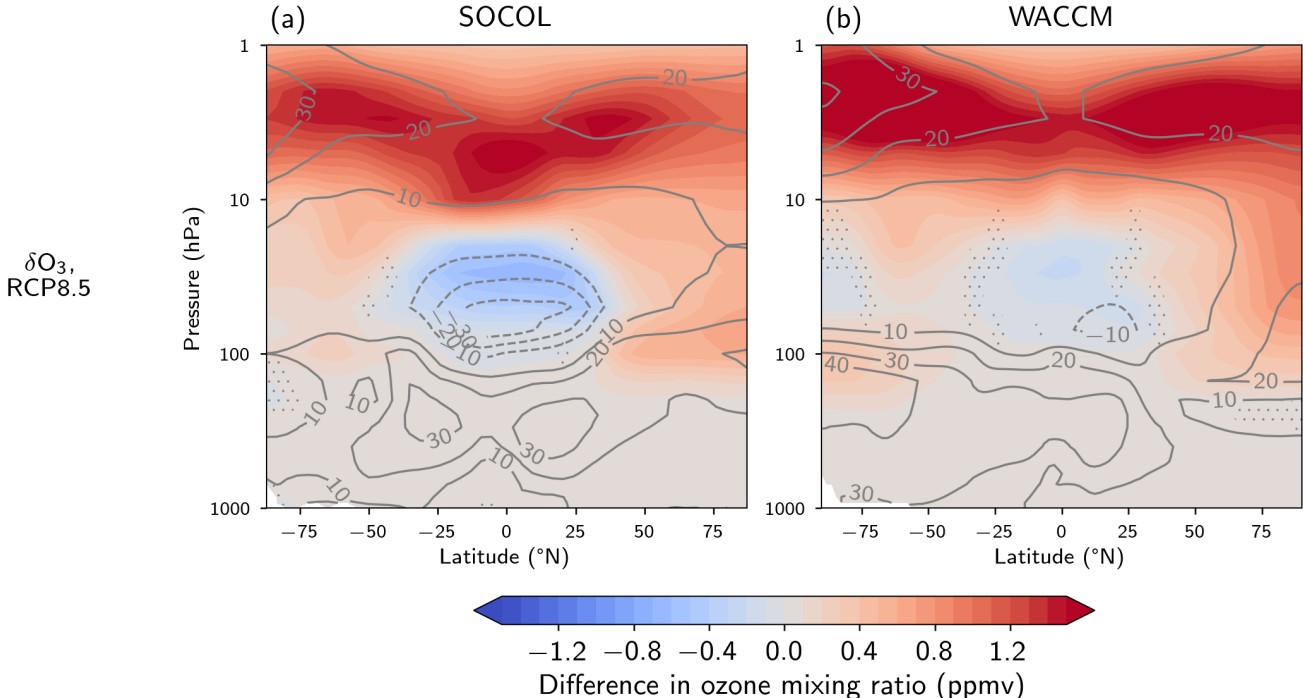

**Figure 2.** Changes in springtime (March-April) zonal mean ozone between the late (2080-2099) and early (2005-2024) 21[st] century under RCP8.5 in SOCOL (a) and WACCM (b). Stippling marks regions where changes are not significant on a 95 % level. Contours show the relative ozone changes in percent.

By contrasting the late period (2080-2099) between the RCP8.5 and RCP8.5_fO$_3$ ensembles, we can unambiguously attribute the impacts of future ozone changes, and we can directly compare these impacts to those projected by 2100 in the RCP8.5 emission scenario. We start by documenting these impacts on temperature and winds on Fig. 3. The temperature response to ozone changes in both CCMs (panels a-b) is broadly coherent with the pattern of ozone changes depicted in Fig. 2, namely (1) a warming of the upper stratosphere by up to 4 K, (2) a cooling of the tropical lower stratosphere by 1 - 3 K, and (3) a warming of the polar stratosphere of up to 4 K in SOCOL and more than 4 K in WACCM. Both models are consistent in the simulated impacts; differences among them in the lower stratosphere are primarily related to differences in the simulated ozone increase rates in these regions. Accordingly, the warming due to ozone increase offsets the radiative cooling by GHGs by approx. 30-35% in the upper stratosphere. In the lower polar stratosphere, the temperature changes due to ozone changes are larger than those simulated under RCP8.5 (Fig. 1). In particular, the warming in the Arctic lower stratosphere (50 - 100 hPa) stands in stark contrast with the total temperature change projected by 2100 (which is close to zero), meaning that the increase in ozone completely compensates any GHG-induced cooling in this region. A similar compensation of radiative effects in this region under springtime conditions has been shown by Kult-Herdin et al. (2023) using idealized time-slice experiments. In the troposphere, we find a weak warming (less than 0.5 K) signal in WACCM: this is likely due to the tropospheric ozone

increases by up to 30% (Fig. 2b). Tropospheric ozone can enhance global warming (Banerjee et al., 2018), but its effects are not robust across the two models and are much smaller than those induced in the stratosphere by stratospheric ozone changes. Hence, future ozone changes under this emission scenario strongly modulate the projected temperature changes in the lower stratosphere, similar to what has been previously reported for abrupt-4xCO2 experiments (Nowack et al., 2015; Chiodo and Polvani, 2019; Li and Newman, 2022).

These changes in lower stratospheric temperature imply a reduction in the meridional temperature gradient. By thermal wind relationship, these changes lead to a weakening of the westerly winds in the polar stratosphere by up to 4 m s$^{-1}$ (Fig. 3c-d): this is statistically significant in both CCMs, with WACCM exhibiting the largest response, consistent with the larger Arctic mid-stratospheric ozone increase in this model (Fig. 2). This signal is in contrast with what is projected by 2100 in both models in the polar vortex in the RCP8.5 ensemble (Fig. 1c-d). which instead shows a strengthening in one model (SOCOL) but no

significant response in another (WACCM). Thus, Arctic ozone offsets the influence of GHGs cooling on the polar vortex (which would by itself strengthen the vortex instead), likely introducing uncertainty in the overall projected changes by 2100 in this region. We note that the signal induced by ozone changes in the NH is much smaller than its SH counterpart during Austral spring (Fig. B3), but it is statistically significant in both models and it is largely driven by shortwave (SW) heating (panels a-b).

We also tested the robustness of the dynamical impacts documented here, by analyzing a set of CCMI-1 runs that isolate the

295 impact of ODS-driven ozone recovery (senC2-fODS2000); these runs were performed by CESM1-WACCM and show a signal that is consistent, indicating weakening of the vortex when ozone recovery is present (Fig. C1). However, the signal is smaller in those experiments, due to the smaller ozone forcing when neglecting any GHG-induced ozone changes (see extended discussion in the Appendix). As springtime is a season with active coupling between the stratosphere and the troposphere (Baldwin and Dunkerton, 2001; Baldwin et al., 2021), we next explore the impacts on downward coupling, with focus on annular modes and

300 the lifetime of the vortex.

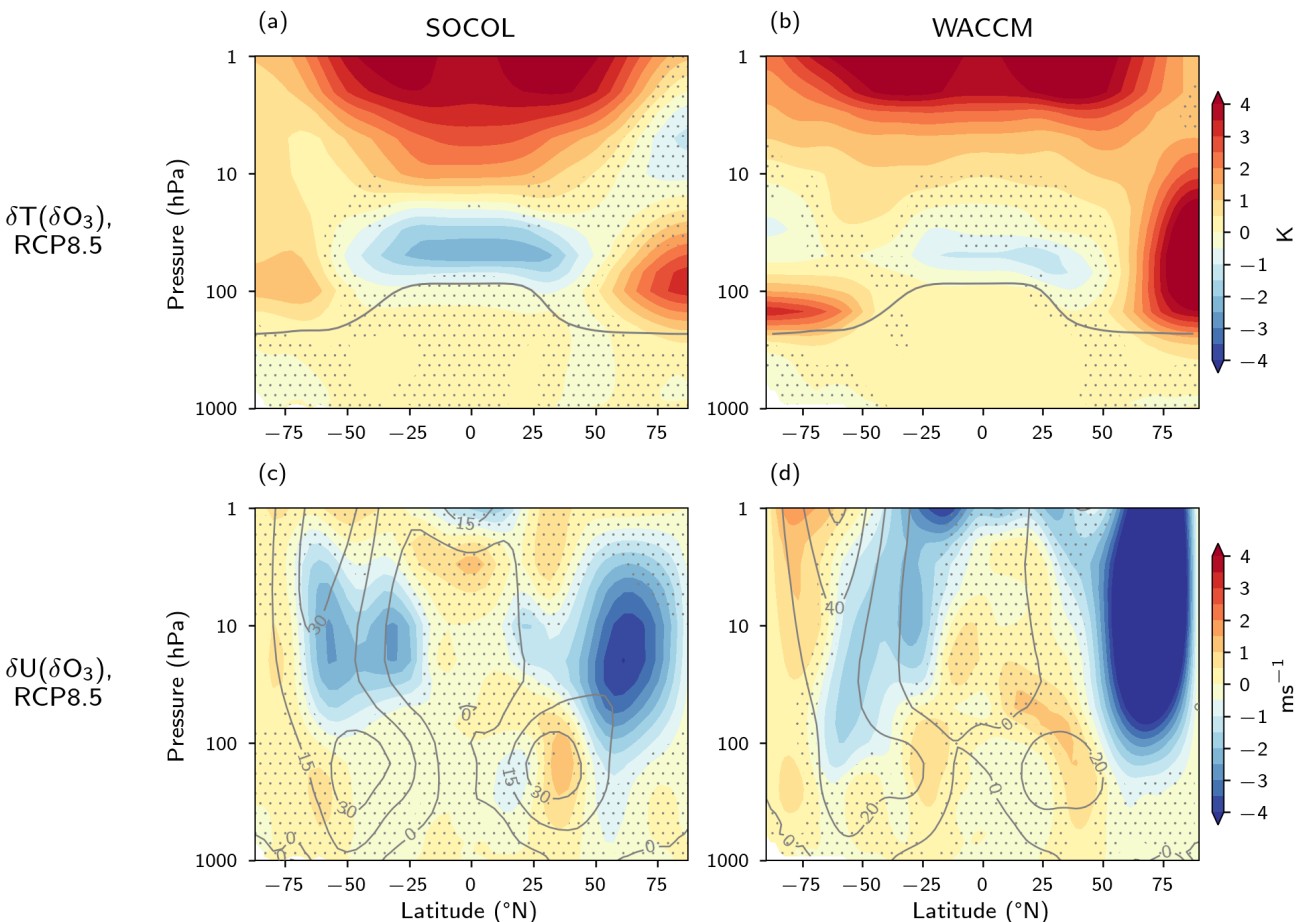

**Figure 3.** Isolated impact of ozone on zonal mean temperature (a, b) and zonal wind (c, d) in springtime (March-April) derived from the difference between RCP8.5 and RCP8.5_fO$_3$ simulations in the late (2080-2099) 21$^{st}$ century in SOCOL (left) and WACCM (right). Contours in a, b show the tropopause height, and contours in c, d depict the climatology of the zonal mean zonal wind (in m s$^{-1}$) in the late 21$^{st}$ century. Stippling marks regions that are not significant on a 95 % level.

### 3.3 Future ozone changes and impacts on stratosphere-troposphere coupling and surface climate

The polar vortex strength is tightly correlated with the stratospheric NAM index (e.g. Baldwin and Dunkerton, 2001), which is usually applied as a measure for the large-scale climate variability in the NH. Here, we utilize the NAM index to quantify changes in the stratospheric mean state, similar to analyses presented in the IPCC (2021) (AR6) for changes in surface NAM. Figure 4 (panel a) displays changes in the springtime stratospheric NAM at 10 hPa over the 21$^{st}$ century in simulations with and without any future ozone changes for individual ensemble members in both models. When only considering the isolated GHG effect (RCP8.5_fO3 simulations), the stratospheric NAM increases compared to its present-day climatology (which is zero by construction) by approximately 0.5 (SOCOL) to 1.0 (WACCM) standard deviations on an ensemble mean. A shift towards a

positive NAM implies a strengthening of the polar vortex. This strengthening is consistent with the GHG-induced warming of the tropical upper troposphere and radiative cooling of the polar lower stratosphere (see the GHG-only impact displayed in Fig. B2), which results in an increased temperature gradient between equator and poles and thus induces a large vertical wind shear. These changes in the stratospheric NAM towards more positive values are evident in all ensemble members, albeit with some uncertainty in the shift's magnitude across members. However, when accounting for the combined effect of GHGs and ozone changes (RCP8.5 simulations), there is considerable uncertainty regarding the sign of stratospheric NAM changes across ensemble members, and changes cancel out in the ensemble mean, consistent with the little robustness of the ensemble mean wind changes displayed in Fig. 1 (panels c-d). Future ozone changes are projected to reduce the GHG-induced shift towards a more positive stratospheric NAM by around -0.5 (SOCOL) to -1.0 (WACCM), cancelling any GHG-induced effects on the stratospheric NAM, effectively leading to no robust changes in the NAM (which is near 0 in the RCP8.5 ensembles). Consequently, the NAM analysis confirms that Arctic ozone increases counteract changes in the polar vortex strength induced by GHGs.

The springtime polar vortex strength is further linked to the occurrence of the FSW. For example, a persistently cold polar vortex in springtime (March and April) tends to be less susceptible to tropospheric wave driving, thus radiatively breaking up late in spring (Waugh et al., 1999). The timing of the FSW has previously been shown to be a key driver of springtime NH climate, as both early and late FSWs are usually associated with a shift of the surface NAM in spring towards its negative phase (Black et al., 2006; Ayarzagüena and Serrano, 2009; Thiéblemont et al., 2019; Butler et al., 2019; Butler and Domeisen, 2021). Similar to changes in the stratospheric NAM, we isolate the impacts of GHGs and future changes in Arctic ozone on the breakup date of the polar vortex in spring. Our analysis reveals that the isolated GHG effect delays the FSW compared to the present-day climatology by approximately 15 (WACCM) to 40 (SOCOL) days, which is consistent with the positive shift in the NAM in the RCP8.5_fO$_3$ ensemble. However, the magnitude of the shift differs remarkably between the two models, which may be attributed to biases in the timing of the FSW under present-day conditions. While both models exhibit a delayed FSW compared to observations, WACCM shows a considerably larger bias (21 days) compared to SOCOL (12 days) (Friedel et al., 2022b). As a result, the impact of GHGs on the timing of the FSW in WACCM may be limited, as they might only be able to induce a small delay in the FSW, before the polar vortex breaks up radiatively. Future changes in Arctic ozone, on the other hand, oppose the GHG-induced delay of the FSW, effectively neutralizing some (SOCOL) or all (WACCM) of the GHG effect (RCP8.5 ensemble). Consequently, the total projected changes (which represent the combined effect of GHGs and future changes in Arctic ozone) of the timing of the FSW manifest differently in the two models. By the end of the 21[st] century, SOCOL projects a delay of approximately 20 days, while WACCM shows no changes. This model discrepancy is consistent with previous findings for CMIP5/6 models (Rao and Garfinkel, 2021), where some models project a delay in the FSW, while others indicate minimal changes (see their Fig. 8). The analysis presented here suggests that the delay in the FSWs would be significantly greater across models — and thus the sign of the projected changes might be more consistent — if it were not for elevated levels of Arctic stratospheric ozone.

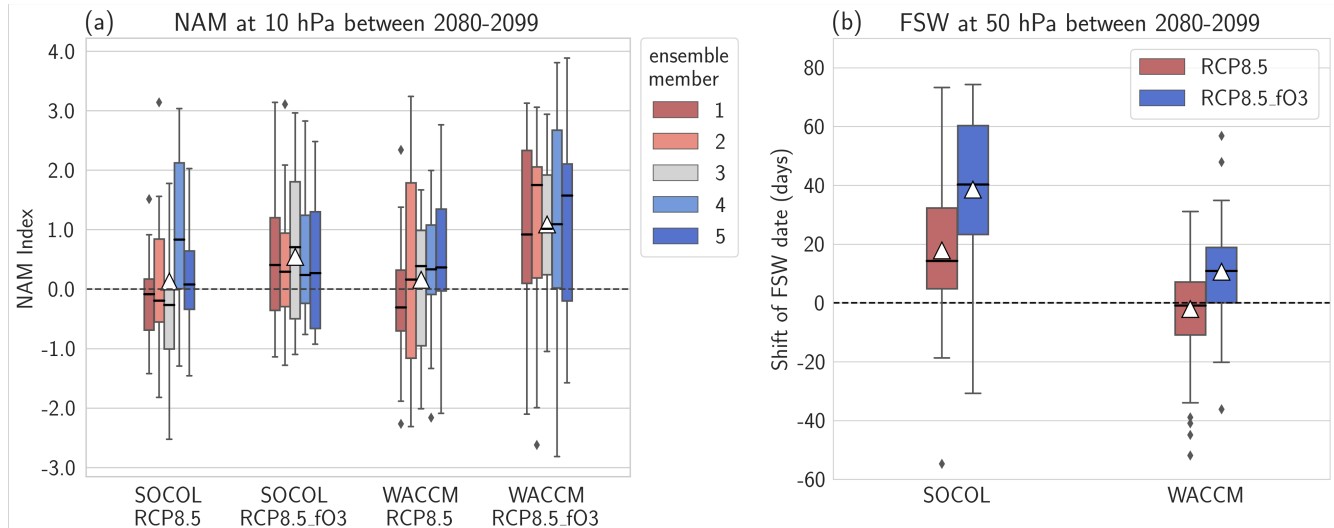

**Figure 4.** (a) Changes in the 10 hPa NAM in springtime (March-April) between the late (2080-2099) and early (2005-2024) 21st century in SOCOL and WACCM RCP8.5 and RCP8.5_fO3 simulations for individual ensemble members. Black lines show the median value for each ensemble members, and the white triangle shows the mean change over all ensemble members. (b) Changes in the SFW date between the late (2080-2099) and early (2005-2024) 21st century in SOCOL and WACCM RCP8.5 and RCP8.5_fO3 simulations. Black lines show the median value, white triangles the mean change. The upper and lower boundaries of the boxes represent the upper and lower quantile of the distribution, and the whiskers show the maximum and minimum values, respectively.

Next, we aim at understanding the mechanisms behind the effects of future ozone changes on the stratospheric temperature and thus on the stratospheric polar vortex. For this purpose, we disentangle the contribution of radiative vs. dynamical heating, by means of offline PORT calculations (see Section 2.3), and display the results in Fig. 5. First, we see that radiative heating
(due to SW absorption by ozone) provides the largest contribution to the ozone-induced stratospheric temperature changes simulated in both models (contrast panels a-b and d-e), explaining almost entirely the warming near the stratopause, and most of the changes in the Arctic stratosphere. Changes in dynamical heating (which are themselves due to changes in upwelling/-downwelling via the BDC) are non-negligible, as they partly offset the radiative cooling from ozone near the tropical lower stratosphere, and contribute to the warming of the polar stratosphere. These temperature changes are the result of a weakening
in the shallow branch of the BDC, as diagnosed via transformed Eularian-Mean (TEM) analysis (Fig. B4 panels c-d), consistent with what has been reported in the context of long-term changes in ozone due to ODSs for the SH polar vortex (Abalos et al., 2019; Polvani et al., 2019). In the Arctic polar vortex, the radiative heating thus slows down the westerlies, allowing for more waves to dissipate (Fig. B4 panels a-b), advancing the breakdown of the vortex (Fig. 4b), inducing further heating of the Arctic stratosphere. These processes are robust across the two models, although their detailed contribution to the overall radiative and
dynamical changes are model dependent, due to differences in the modeled structure and amplitude of the ozone changes.

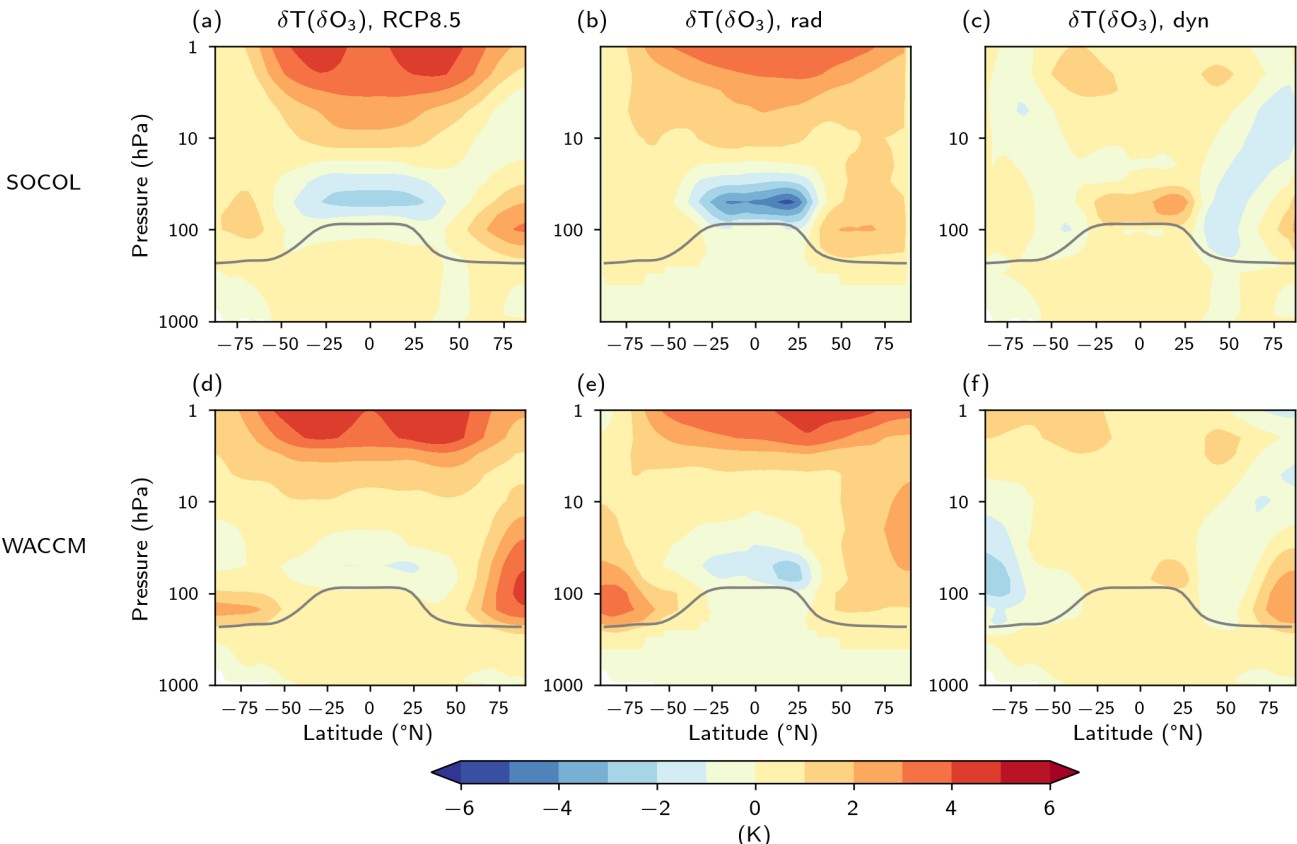

**Figure 5.** Separation of dynamical and radiative contributions to the total springtime temperature anomalies induced by ozone in the two models. Panels (a) and (d) show the total ozone-induced temperature changes simulated by SOCOL and WACCM (note that we plot the same panels as in Fig.3 a-b). Panels (b-f) show the contribution of radiative (b,e) vs. dynamical (c,f) heating on springtime stratospheric temperatures changes. The radiative contribution is computed using the fixed dynamical heating (FDH) assumption, calculating the ozone-induced radiative flux changes, but neglecting any temperature changes due to large-scale circulation (see methods). The dynamical heating contribution (panels c and f) is obtained by subtracting the "radiative" (panels b and e) from the "total" temperature response (panels a-d).

Changes in the polar vortex due to GHGs and ozone can potentially affect tropospheric and surface climate: these are examined next. We first explore the response of tropospheric and surface climate during springtime, by documenting the projected sea-level pressure (SLP) response in Fig. 6 in the ensembles including future ozone changes (panels a, d) and in the ensembles without future ozone changes (panels b, e). Interestingly, we see that the total projected changes in springtime by 2100 under RCP8.5 are largely model dependent, with SLP declining in SOCOL over the Aleutian sector (panel a), and increasing across the North Pacific in WACCM (panel d). We find a much clearer positive Arctic Oscillation (AO) pattern in WACCM (which is similar to other CMIP6 models) in the ensemble excluding the effects of any future ozone changes (panel e). Most remarkably, when isolating the effects of ozone changes (by taking differences between RCP8.5 and RCP8.5_fO$_3$ - panels

c, f), we find a much more consistent response across both models, reminiscent of a negative AO pattern, with positive SLP
anomalies over Greenland and the Arctic ocean. Note that such surface signal is highly consistent with a weak stratospheric
polar vortex (Fig. 3), being reminiscent of what occurs in the aftermath of Sudden Stratospheric Warmings (SSWs) (Baldwin
et al., 2021; Domeisen and Butler, 2020). However, the SLP signal induced by future ozone changes (panel c, f) is much smaller
- by approximately a factor of two - than that induced by GHGs (panels b, e) and is on the fringe of statistical significance (i.e.,
only a limited region around Greenland is significant at the 95 % level).

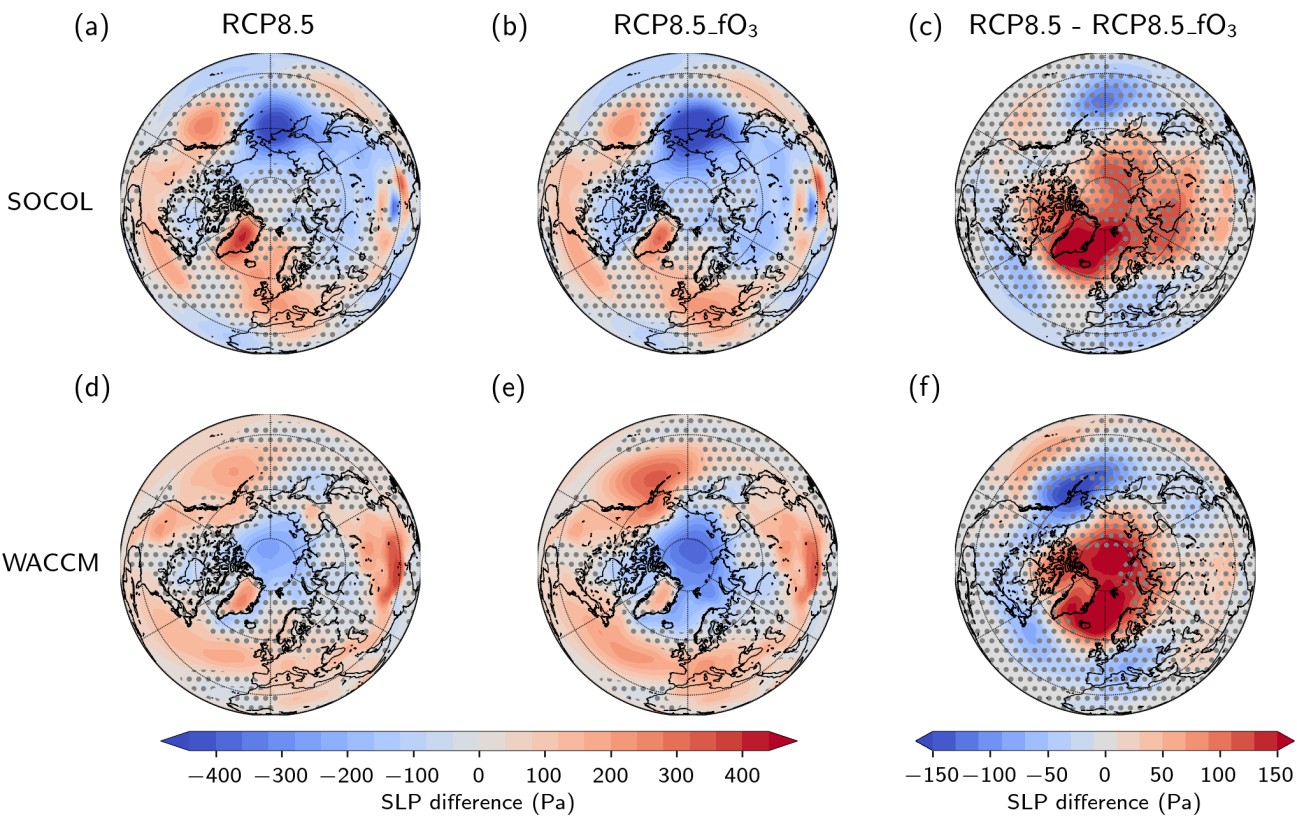

**Figure 6.** Changes of Sea Level Pressure (SLP) in springtime (March-April) due to the combined effect of GHGs and long-term ozone
changes simulated under RCP8.5 (a, d), isolated changes of SLP due to GHGs alone (b, e), and isolated changes of SLP due to future
changes in ozone (RCP8.5 minus RCP8.5_O3, c,f) between the late (2080-2099) and early (2005-2024) 21$^{st}$ century under this scenario in
SOCOL (upper row) and WACCM (lower row). Stippling marks regions where changes are not significant on a 95 % level.

These effects on SLP also have implications for tropospheric (500 hPa) zonal wind, as shown in Fig. 7. Similar to SLP,
we see that the overall tropospheric zonal wind changes projected by 2100 are very different across the two models (panels
a, d). In SOCOL, we see a significant strengthening of the westerly winds across the North Atlantic and over Eurasia in both
ensembles (panels a-b), whereas in WACCM, we find an acceleration of westerlies on the poleward side of the jet maximum

on the Western North Atlantic in the ensemble without any ozone changes (panel e), similar to other CMIP5 models (Barnes
and Polvani, 2013). By quantifying the effects of ozone changes (panels c, f), we again gain a more consistent picture from
both models, indicating a weakening of tropospheric westerly winds on the poleward side of the eddy-driven jet over the North
Atlantic by up to 1 m s$^{-1}$. Again, the ozone-induced signals are much smaller (and of opposite sign) than those induced
by GHGs (compare panels c, f against panels b, e), but are much more robust across the two models. Furthermore, they are
consistent with observational and modeling evidence for the effects of a disturbed stratospheric polar vortex on the Atlantic
storm tracks (Afargan-Gerstman and Domeisen, 2020). Thus, ozone provides yet another mechanism in the "tug of war" of
processes opposing the influences of climate change induced by GHGs (Shaw et al., 2016).

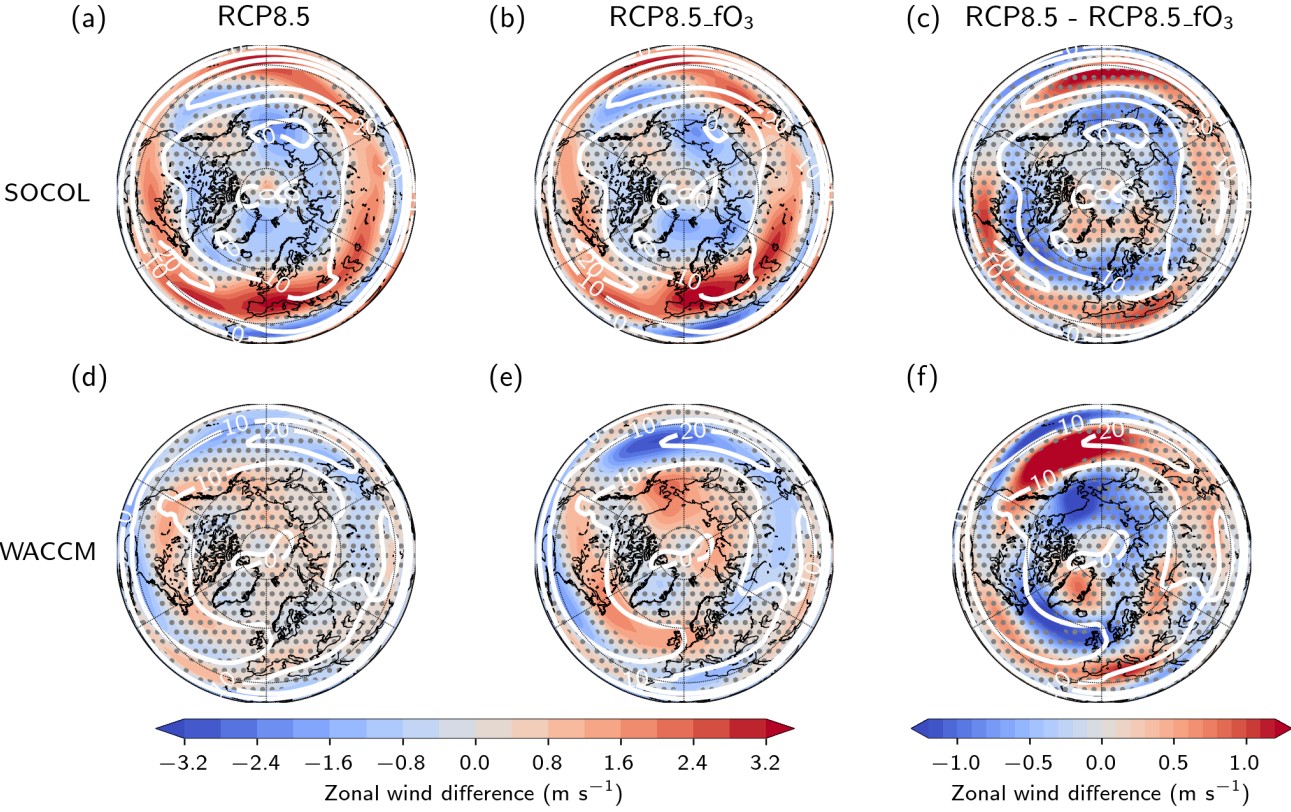

**Figure 7.** As in Fig. 6 but for zonal wind at 500 hPa. Contours show the climatology of the mean zonal wind in the early 21$^{st}$ century (first
20 years of simulation) in SOCOL and WACCM, respectively.

## 4   Conclusions

In this study, we have explored and quantified the impacts of future springtime Arctic ozone changes on future projections in
the NH for the stratospheric circulation and surface climate. Our experimental setup allows us to assess the effects of future

ozone changes and GHGs under a high-emission scenario from IPCC-AR5 (RCP8.5, see Meinshausen et al. (2011)). The key findings are summarized as follows:

- Future projections of the large-scale atmospheric circulation response to climate change in the NH are widely different across the two CCMs examined here, similar to recent inter-model comparisons. For example, the stratospheric polar vortex strengthens by 4-6 m s$^{-1}$ and breaks up later (by 15 days) in SOCOL, while no changes in any of the vortex
diagnostics are simulated in WACCM. Similarly, the projected changes in tropospheric and surface climate differ across the two models.

- Future changes in GHGs and ODSs drive sizable increases (by up to 20%) in springtime ozone in the Arctic. These changes in ozone lead to warming of the Arctic stratosphere and consequently, weaken the polar vortex, advancing its break-up date by 10-20 days. The warming of the polar stratosphere is primarily due to enhanced solar absorption by
395 Arctic ozone abundances, and partly due to enhanced downwelling (from enhanced wave dissipation).

- In the troposphere, we find regional signals that are coherent with those induced in the stratospheric polar vortex. Namely, an increase in Arctic stratospheric ozone abundances leads to a negative Arctic Oscillation pattern and an equatorward shift of the eddy-driven jet over the North Atlantic. While the effects in the stratosphere are detectable, those in the troposphere are only on the fringe of significance, although they are very robust across the two models used in this study.

Taken together, these results suggest that future trends in ozone oppose the effects of GHGs. In the NH polar stratosphere, these ozone changes almost entirely cancel the effects of GHGs (by up to 50-100% in metrics such as the NAM and vortex breakup date), while in tropospheric and surface climate, ozone changes only partly counteracts them (by up to 10-20%). In so doing, springtime Arctic ozone increase shapes the large-scale atmospheric circulation response to climate change in the NH. Hence, the uncertainty in the overall impacts of climate change on certain dynamical metrics, such as the stratospheric
polar vortex strength and timing of its break-up, partly result from the competition between ozone and GHGs. These effects are similar in the two CCMs examined here, despite the large differences in the models' projections of global warming. We acknowledge that the large and previously unrecognized role of Arctic ozone may partly be due to the large ozone changes arising from the high-emission scenario considered in this study (RCP8.5) (WMO, 2022). The large ozone changes in this scenario are partly due to the large CH$_4$ forcing, although large abundances of N$_2$O would reduce the contribution of non-CO$_2$
GHGs to the projected ozone changes in this specific scenario. Assuming linearity of the dynamical response to the ozone changes, the forcing by future Arctic ozone trends in this scenario is larger than by past ozone depletion trends (1960-2000), for which a dynamical impact has been ruled out in several Ozone Assessments (see WMO, 2014, 2018, 2022). Moreover, our study only highlights the impacts during springtime, for which the forcing (and thus dynamical impacts) is the largest.

In spite of these caveats, our results demonstrate, for the first time, that long-term changes in the ozone layer can signif-
415 icantly influence the circulation response in the NH. We obtain this conclusion in the context of a high-emission, but yet policy-relevant, IPCC scenario, thereby expanding on previous results on the SH (Mindlin et al., 2021; Ivanciu et al., 2022) and idealized experiments using abrupt-4xCO2 (Li and Newman, 2022; Chiodo and Polvani, 2019). The implications of this

study are threefold. First, models need to include a consistent representation of stratospheric ozone trends; models without interactive chemistry thus need to impose a forcing-consistent ozone data-set (Maycock, 2016). Second, in models with interactive chemistry (CCMs), any uncertainty in the projected recovery rates could result in uncertainty in the degree to which ozone offsets the effects of GHGs in these types of models. Third, models may exhibit different sensitivities to long-term ozone trends (e.g., Lin et al., 2017). Considering these aspects and in particular constraining any of these potential sources of uncertainty might help to decrease the longstanding uncertainty about the dynamical effect of a given radiative forcing (see examples of this e.g. for the polar vortex in Karpechko et al. (2022)). Given the sizable effects of the stratosphere on tropospheric and surface climate (Simpson et al., 2018), an improved understanding of the role of Arctic ozone as a driver could benefit regional climate change projections in springtime, especially in the mid- and high latitudes, where the uncertainty in the projections is sizable (see IPCC, 2021, Fig. 4.30).

*Data availability.* Data for the transient simulations for WACCM and SOCOL-MPIOM, as well as all scripts used for the analysis in this study are available upon request.

*Code and data availability.* The code needed to produce the plots and the analysis can be made available by the corresponding author upon reasonable request.

## Appendix A: Coherence between vortex shape and imposed ozone climatology

We verify that imposing an ozone climatology in the RCP8.5_fO$_3$ ensemble does not introduce any major inconsistencies with the vortex morphology, by contrasting the 3-D daily ozone climatology being used in the RCP8.5_fO$_3$ and the vortex location (measured by geopotential height values at the 50 hPa level), under both present-day (2005-2024) and future (2080-2099) conditions in the RCP8.5_fO$_3$ ensemble. Fig. A1 shows the location and shape of the polar vortex, along with the ozone climatology being used. As the Figure shows, the vortex differs between the two models, being stronger and less elongated in WACCM than in SOCOL-MPIOM. Most importantly, the morphology of the vortex does not change strongly towards the end of the 21st century in any of the two models (contrast left vs right panels). Thus, the minimum in the ozone climatology being imposed in these ensembles (color shading) approximately coincides with the location of the polar vortex. Similarly, the higher ozone abundances correspond to the edge of the polar vortex in both present-day and also future conditions. From this, we conclude that the "mismatch" between future dynamics and present ozone field mentioned is unlikely to affect the results, and any differences between the two ensembles are due to long-term ozone changes, and not due to potential artifacts.

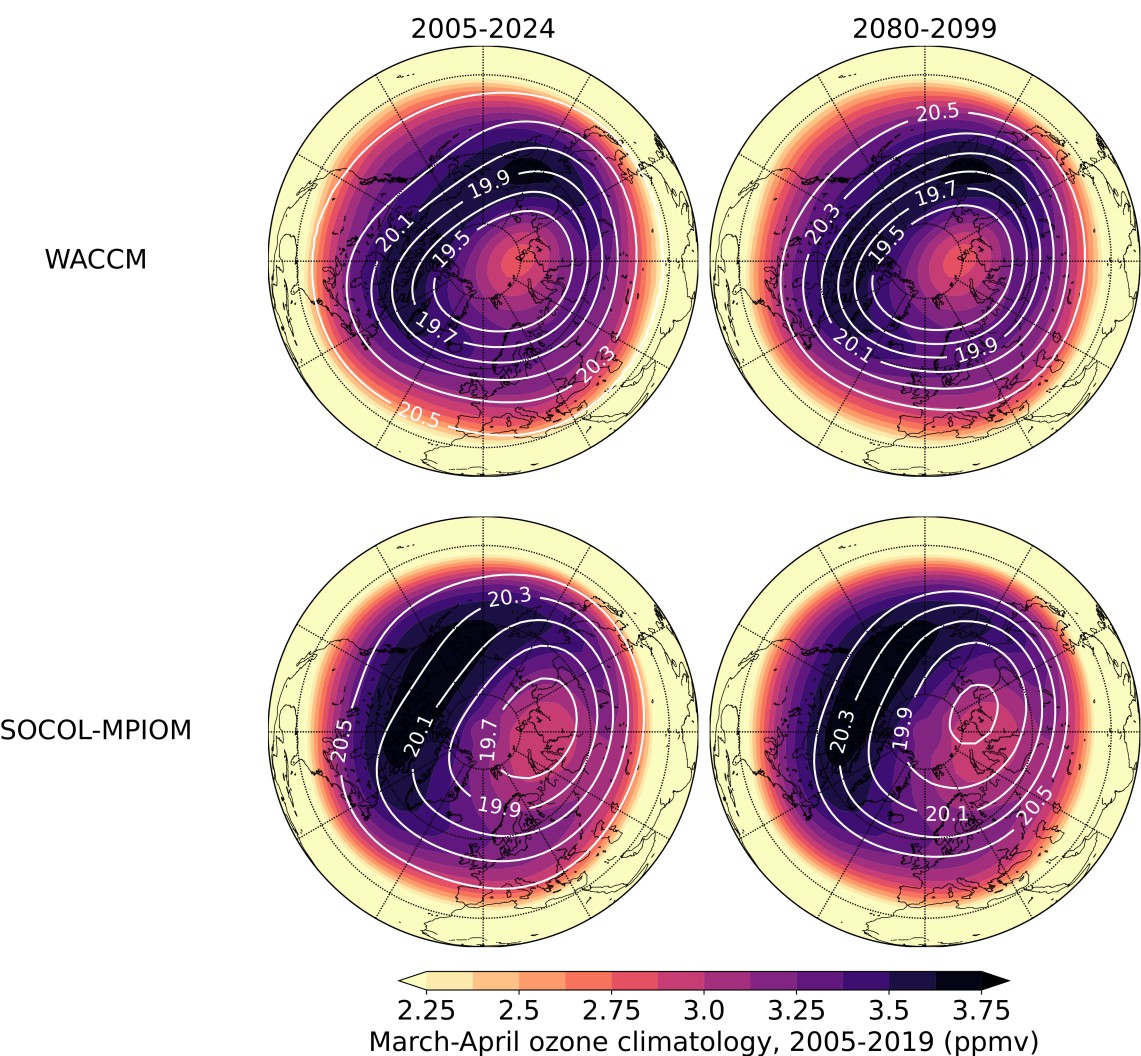

**Figure A1.** Springtime ozone climatology at 50 hPa used for the RCP8.5_fO₃ experiment (color shading), with superimposed geopotential height (in units of km) corresponding to the same pressure level (isolines) for WACCM (top) and SOCOL-MPIOM (bottom), averaged over the "early" (2005-2024; left) vs "late" period (2080-2099; right). The lowest isolines (19.5-19.7 km) identify the approximate location of the stratospheric polar vortex, while the meridional gradient in geopotential heigh corresponds to its strength. As one can see, the lowest ozone concentrations that are imposed in the fO₃ ensemble correspond to the location of the vortex in both time periods.

## Appendix B: Global warming and changes in the stratosphere projected in the high-emission scenario

In the following, we show the projected changes in key metrics of global warming, namely global mean surface temperature, and tropical average residual upwelling ($w^*$) in both models (Fig. B1). Both models exhibit warming of 0.3 - 0.5 K decade$^{-1}$, and a strengthening of the tropical upwelling by 3-4 % decade$^{-1}$, in line with other high-top climate models (Butchart, 2014). To analyze the isolated impact of GHGs alone (i.e., without concomitant future ozone changes) on zonal mean temperature, we display the changes in this field from the RCP8.5_fO$_3$ in Fig. B2.

To examine the seasonality of the projected climate change in the lower stratosphere (resulting from both GHGs and ODS reduction), we then plot the change in four variables (shortwave heating, residual upwelling, temperature and zonal mean zonal wind) as a function of month in Fig. B3. Here, we see that the projected strengthening of tropical upwelling is balanced by a strengthening in the downwelling at mid and high-latitudes (panels c, d). This response is consistent with an acceleration of the shallow branch of the BDC throughout the year in both hemispheres, resulting from a strengthening of the subtropical jets

(panels g-h). In the Arctic stratosphere, no significant temperature changes are projected, except for the fall season (cooling in panels e-f). The isolated effect of the recovery of the ozone layer (Fig. B4) leads to warming of up to 8 K in the SH high latitudes during austral spring and early summer (September – December) and a weakening (and earlier breakdown) of the SH polar vortex, consistent with the vast literature on the subject (e.g., Ivanciu et al., 2022). The ozone increase also leads to significant warming by 2-4 K in the Arctic boreal spring (panels e-f), and a weakening of the westerlies at 60-70N (panels

g-h). These signals are much smaller than those in SH, which are associated with the closing of the Antarctic ozone hole, but they are detectable in both models. These changes in the westerlies have repercussions on wave propagation, and in turn on the overturning circulation in the stratosphere (Fig. B5). More specifically, ozone-induced changes in the meridional temperature gradient near the tropopause weaken the zonal mean zonal winds in the sub-polar lower stratosphere, resulting in more wave dissipation in the Arctic stratosphere, although the exact location of the enhanced wave dissipation depends on the model

(panels a-b). These changes in wave dissipation in turn induce stronger downwelling in the polar lower stratosphere at 30 - 100 hPa), although the details of this structure are again model-dependent, with WACCM exhibiting the largest downwelling anomalies (and thus also largest dynamical heating) (panel d).

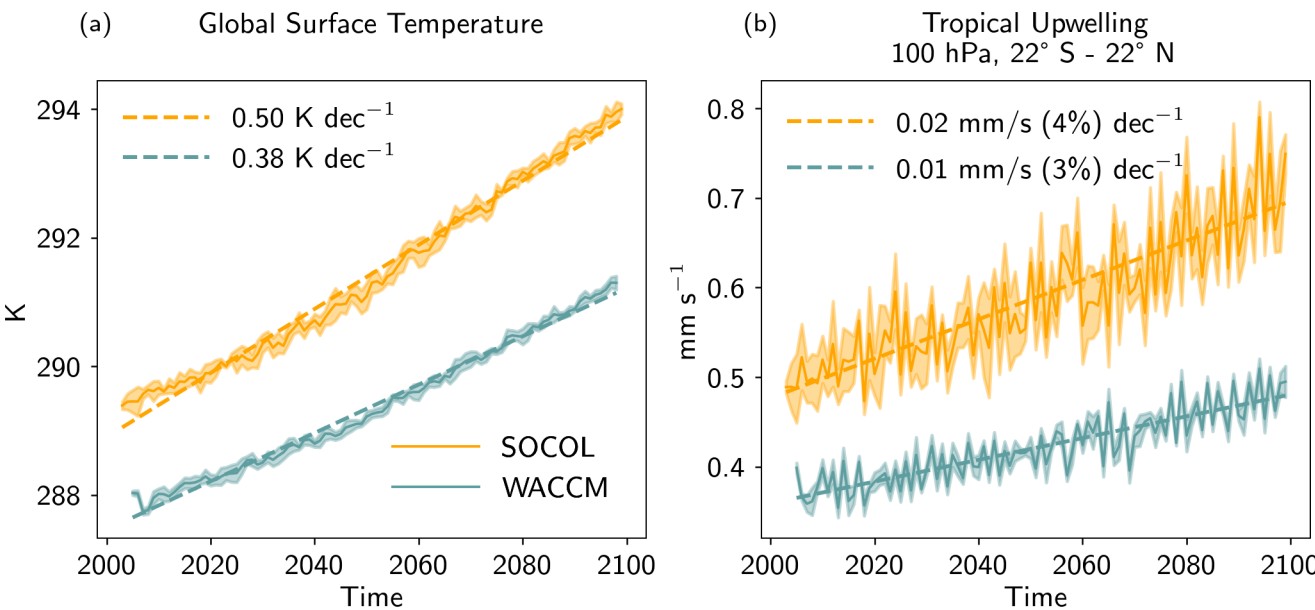

**Figure B1.** Annual mean global mean changes in surface temperature in WACCM and SOCOL in the RCP8.5 ensemble (a), and tropical average residual upwelling (w*) (b). The shading represents the uncertainty across the five ensemble realizations performed for each model.

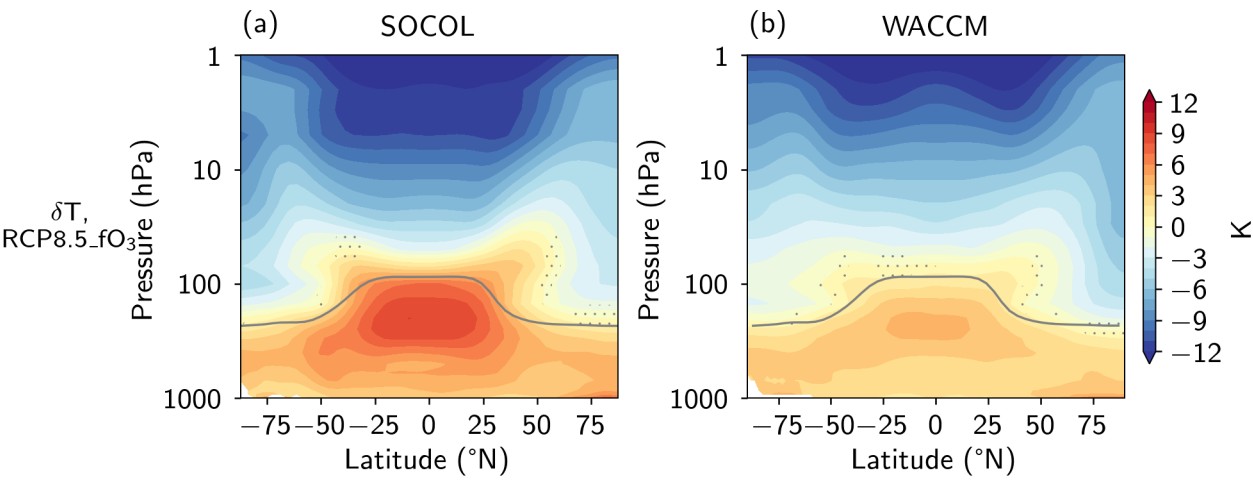

**Figure B2.** Zonal mean temperature changes in springtime in SOCOL (a) and WACCM (b), calculated as difference between late (2080-2099) and early (2005-2024) period in the ensemble with fixed present-day ozone climatology (RCP8.5_fO$_3$). Stippling masks changes that are not significant at the 95% level. Units K.

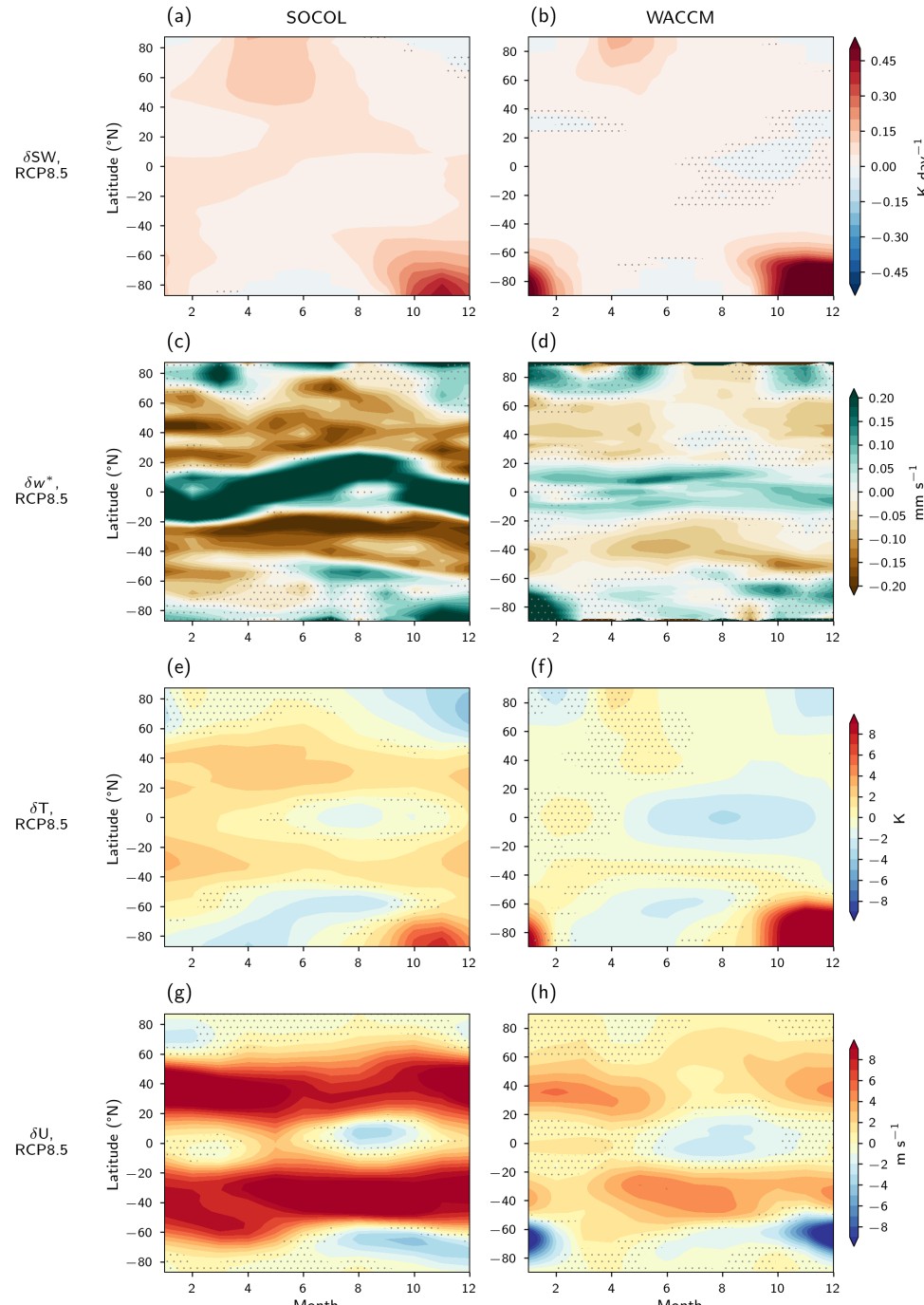

**Figure B3.** Projected changes in the RCP8.5 ensemble, in zonally averaged SW heating rates (panels a and b), residual upwelling (panels c and d), temperature (panels e and f), zonal wind (panels g and h) for each month at the 70 hPa pressure level. Stippling masks changes that are not significant at the 95 % level.

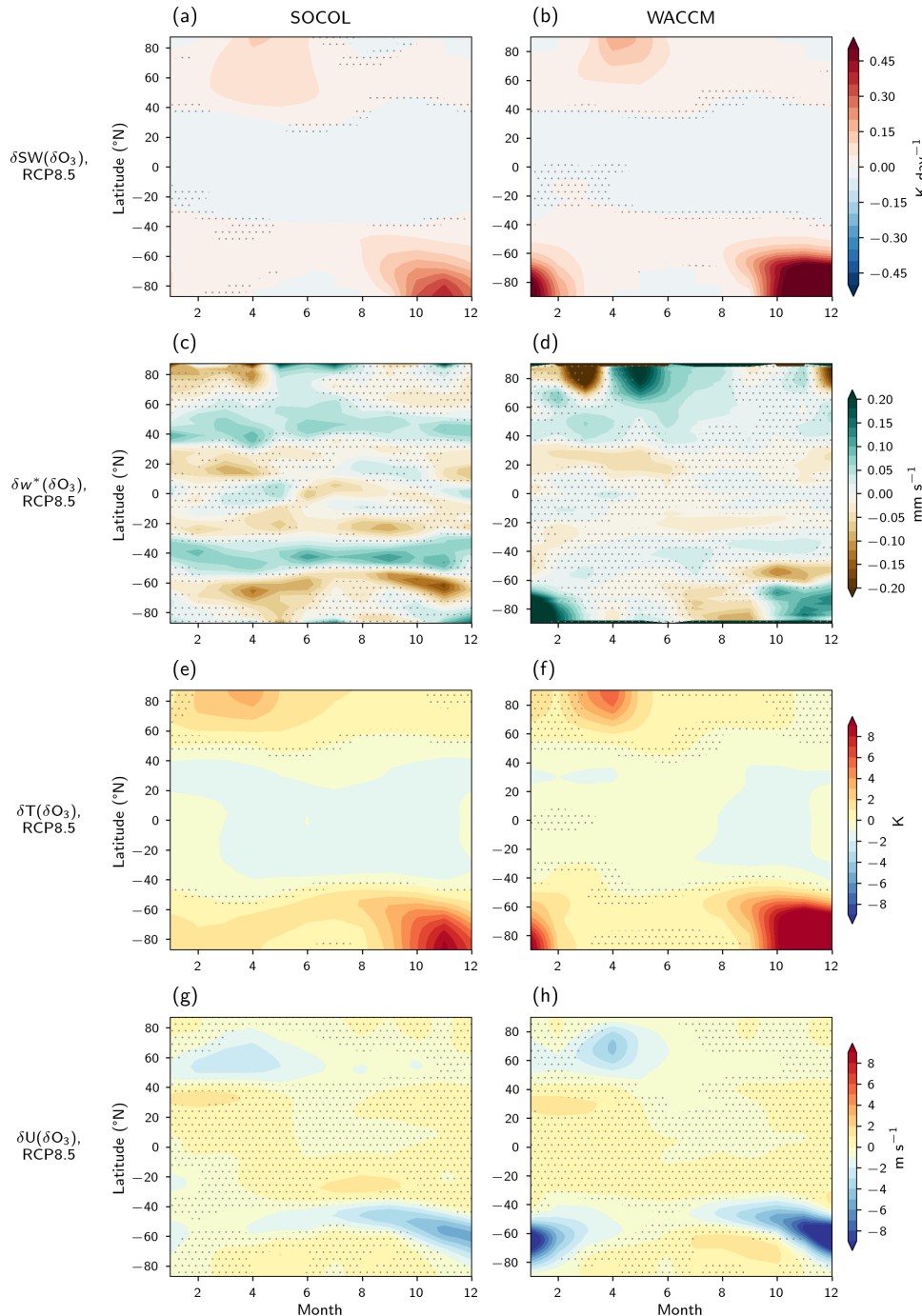

**Figure B4.** As Fig. A2, but isolating the contribution by future changes in Arctic ozone, quantified as difference between the RCP8.5 and RCP8.5_fO$_3$ ensembles in the late period (2080-2099). Stippling masks changes that are not significant at the 95 % level.

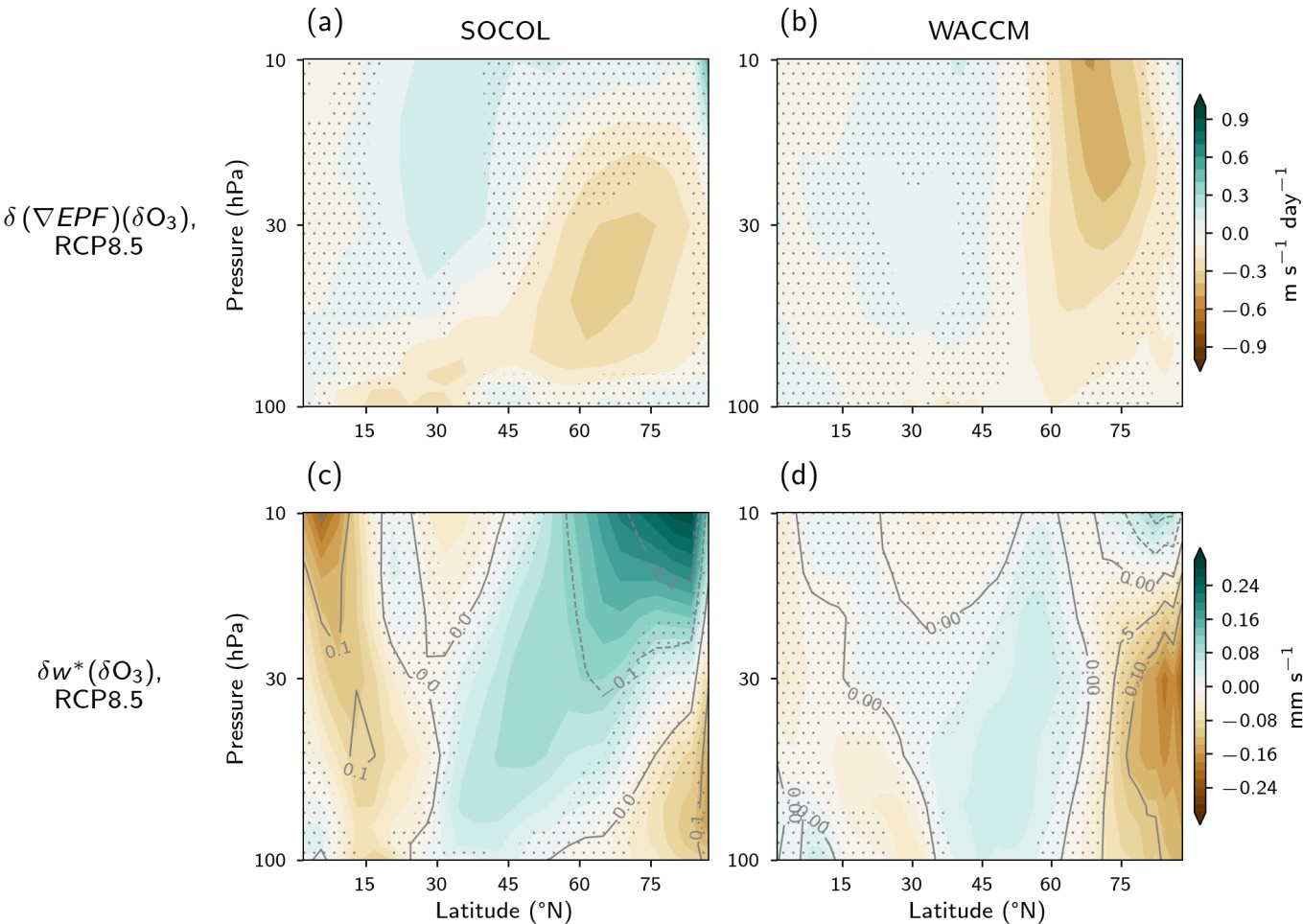

**Figure B5.** Isolated impact of Arctic ozone changes (quantified as difference the RCP8.5 and RCP8.5_fO₃ ensembles in the late period of 2080-2099) during springtime (March – April) on EP flux divergence (a, b) and residual vertical velocity (c, d) in SOCOL (left column) and WACCM (right column). Contours in c and d show the dynamical heating rate in K day$^{-1}$. Residual velocities and EP flux divergence are calculated using the Eularian mean framework according to Andrews et al. (1987). For details on the calculation of dynamical heating rates, please refer to Friedel et al. (2022a). Changes that are not significant at the 95% level are stippled.

## Appendix C: Impact of ODS on the circulation based on CCMI-1 experiments

Lastly, we examined the impact of ODS-induced ozone changes on the polar vortex, by analyzing the CCMI-1 sensitivity experiments performed by CESM1-WACCM; results for zonal mean zonal wind are shown in Fig. C1. It is readily evident how ODS emissions, via their effects on ozone, induce a weakening of the polar vortex. The sign of the changes is consistent with what is reported in this paper for the Arctic ozone under RCP8.5 (Fig. 3), but the magnitude is much smaller (by 50%), most likely due to the smaller ozone forcing in these runs.

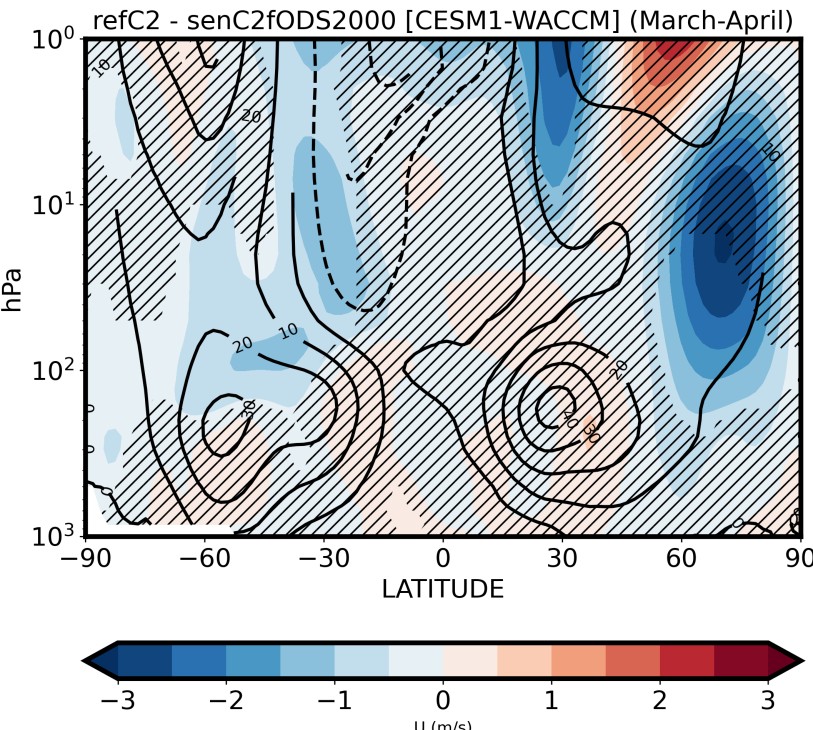

**Figure C1.** Isolated impact of future ODS emissions (and their induced ozone changes) on zonal mean zonal wind from CESM1-WACCM runs contributing to CCMI-1, calculated as springtime (March - April) ensemble mean differences between three reference runs ("refC2" in Morgenstern et al. (2017)) and three future scenarios with ODS surface-mixing ratios fixed at their year-2000 values ("SEN-C2-fODS2000" in Morgenstern et al. (2017)). Contour lines show the climatological averages over 2005-2025 in the "refC2" ensemble. Changes that are stippled are not significant at the 95% level. Units m/s.

*Author contributions.* G.C., M.F. and S.S. contributed equally to this publication. G.C., M.F., S.S. and A.S. performed and processed the modelling experiments, and analysed the output. S.S., M.F., G.C. interpreted the data, with input from D.D., T.S. and F.Z. G.C. wrote the paper with input from all authors.

*Competing interests.* The authors declare no competing interests.

*Acknowledgements.* G.C. and M.F. gratefully acknowledge the Swiss National Science Foundation for support via the Ambizione Grant N. PZ00P2_180043. Support from the Swiss National Science Foundation through project PP00P2_198896 to D.D. is gratefully acknowledged. We thank the anonymous reviewers #1 and #3 for their insightful comments and suggestions which helped to improve the manuscript substantially.

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
