# Peer review of "The influence of future changes in springtime Arctic ozone on stratospheric and surface climate"

_EGUsphere, 2023_

## Author Response (AR1)

**Reply to reviews: The influence of springtime Arctic ozone recovery on stratospheric and surface climate**

Gabriel Chiodo, Marina Friedel, Svenja Seeber, Daniela Domeisen,
Andrea Stenke, Timofei Sukhodolov, Franziska Zilker

July 29, 2023

RC = Reviewer Comment
AR = Author Reply

**General remarks to the editor and referees**

We thank all the reviewers for their helpful comments: we have addressed all of them. One major issue raised by all reviewers refers to the recovery definition, according to which our definition differs from the canonical definition employed in other papers and WMO Assessments. We consider the effects of the total long-term ozone changes in a high-emission scenario (RCP8.5), rather than those due to ODS emissions alone. Our paper aims at exploring and documenting the dynamical impacts of projected long-term ozone changes in the Arctic and global stratosphere and so by definition, it contains the contribution of all major GHG and ODS emissions to the ozone changes, irrespective of the individual emissions from each of these separate sources. We realize that this can lead to confusion and have therefore changed any reference to "Arctic ozone recovery" to "future changes in Arctic ozone" throughout the manuscript, as well as in the title, which is now *The influence of future changes in springtime Arctic ozone on stratospheric and surface climate* (as suggested by Ref#3). Moreover, we have now included a short discussion of the role on the interactivity of ozone, and show that our results are not due to any dynamical inconsistencies or artifacts created by prescribing an ozone field (rather than interactively simulating one) in the ensemble with fixed present-day ozone values. Instead, the dynamical changes in stratospheric and surface climate reported in our paper are unambiguously driven by long-term Arctic and global stratospheric ozone changes. For more details, please see our replies below.

**Reviewer 2**

**RC 2.1**   I will comment quickly only on my point 1. It is simply inappropriate to define "recovery" any way you like, without regard to well established conventions in the field by now, as employed in the international WMO assessment process and many previous papers. This should be changed throughout if this paper is to be publishable, and the paper should also conduct runs that make clear what drives the conclusions: CFCs? I doubt it in 2100. CH4? N2O? CO2? These each have vastly different implications.

**AR 2.1**   We thank the reviewer for their response. We agree that the usage of a different "ozone recovery" definition compared to the WMO Assessments and other studies might be misleading, and have now changed the title of the manuscript to "The influence of future changes in springtime Arctic ozone on stratospheric and surface climate". Furthermore, we have replaced the term "ozone recovery" with "future changes in Arctic ozone" throughout the manuscript.

We appreciate the reviewer's suggestion concerning the single forcing experiments, to better attribute the ozone changes and the resulting dynamical impacts. However, we wish to note that our paper (as stated in the abstract and conclusions) aims at investigating the climatic impacts from the NH long-term ozone changes themselves rather than from specific emissions and processes that drive the ozone evolution. In other words, we are less interested in the individual drivers of the long-term ozone changes, but rather in the resulting dynamical impacts from these ozone changes: this has never been quantified for the Arctic and NH and is the main novelty of this present study. For this purpose, we use a RCP8.5 scenario, which integrates the effects of all future major GHG forcings in a high-emission scenario.

Quantifying the impacts of individual emissions, as suggested by the Referee, is interesting in its own right from a policy perspective, but it would be out of the scope of the present study. Using simple arguments assuming linearity, one could attribute the dynamical changes shown in this paper to the main drivers of long-term ozone changes in a future scenario like RCP8.5. Projected increases in Arctic ozone tend to be equally driven by declining ODSs and increasing GHGs (see S. S. Dhomse et al., 2018 - see their Fig. 3b), with GHGs becoming more dominant in high-emission scenarios (see A. Butler et al., 2016). Even then, the contribution of ODS (CFCs) is not negligible and possibly also the resulting dynamical changes are partly due to CFCs. To come to this conclusion, we have performed an additional analysis of existing CCMI-1 experiments from CESM1-WACCM, comparing a sensitivity scenario using fixed ODS at year 2000 ("SEN-C2-fODS2000" in Morgenstern et al., 2017) with a reference future scenario with time-varying ODSs ("refC2" in Morgenstern et al., 2017). We found a weakening of the Arctic polar vortex due to declining ODSs, which is consistent with finding from our paper (see Fig. R3). The magnitude of the ODS-only signal derived this way is smaller (around 50%) compared to the signal presented in the

manuscript (Fig. 3 c-d in the main manuscript), consistent with the smaller forcing in this case (ODS-driven ozone changes are roughly half of the overall changes in Arctic stratospheric ozone projected in high-emission scenarios scenario - see S. S. Dhomse et al., 2018). This further proves that ODSs are, indeed, an important driver of long-term ozone changes and the resulting dynamical impacts on the Arctic stratosphere reported in this study.

We have included a discussion of this figure and more generally about the contribution of ODS emissions alone in the main text (L295) and in the Appendix. We have also added a note about our focus on the impacts of the overall ozone changes, rather than individual emissions, as follows: *Our paper aims at documenting the dynamical impacts of projected long-term ozone changes in the Arctic and global stratosphere and so, by definition, it contains the contribution of all major GHG and ODS emissions to the ozone changes, irrespective of the individual emissions from each of these separate sources.*

**Reviewer 3**

**RC 3.1**   The paper by Chiodo et al documents climate changes that can be attributed to future increases in Arctic stratospheric ozone due to removal of ozone depleted substances and increase in greenhouse gas (GHG) concentrations. They show that projected Arctic ozone increase in spring, if happens in isolation from GHG increases, would lead to a warmer Arctic stratosphere, weaker polar vortex and more negative Arctic oscillations. While GHG induced changes will dominate, the ozone induced changes are nevertheless sizable and cancel out some of GHG induced changes. The authors used two independent atmosphere-ocean models to demonstrate that the results are robust across models.

The paper is well-written and well-illustrated. I find the results interesting and potentially publishable. My major concerns are with the methodology and interpretation of the results as explained below:

Major comments:

1. The authors mix simulations with interactive and prescribed ozone but attribute the results to the differences in ozone, claiming that the different methodology does not have an impact on the results. I understand that the prescribed present day 3D ozone results in similar present day climate as the interactive ozone (as per discussion in lines 118-124). This is expected because applied present-day 3D climatological ozone is roughly coincident with present-day dynamical fields. However, as the dynamics will change due to GHG increases there will be mismatch between future dynamics and present ozone field, which will likely affect the results. I do not know how large the effect will be, however this surely needs to be considered as a caveat and its implications discussed.

**AR 3.1**  We thank the reviewer for their comment. We have now included Figure R1 in this letter, to support the statements around L118 concerning the similarity between interactive and prescribed ozone runs under present-day conditions, reaffirming the validity of the prescribed ozone approach in the "RCP8.5_fO$_3$" ensemble.

As the reviewer points out, this inconsistency may become bigger and more important in the future, as the vortex changes in response to GHGs. One possible inconsistency concerns the location of the ozone minima and the vortex shape. We investigated this potential caveat, by contrasting the 3-D daily ozone climatology being used in the RCP8.5_fO$_3$ and the vortex location (measured by geopotential height values at the 50 hPa level), under both present-day (2005-2024) and future (2080-2099) conditions in the RCP8.5_fO$_3$ ensemble. As shown in Fig. R2, the location and shape of the polar vortex differ between the two models, consistent with other CMIP6 studies (e.g. Rao and Garfinkel, 2021), being stronger and less elongated in WACCM than in SOCOL-MPIOM. Most importantly, the morphology of the vortex does not change strongly towards the end of the 21st century in any of the two models (contrast left vs right panels). Thus, the minimum in the ozone climatology being imposed in these ensembles (color shading) approximately coincides with the location of the polar vortex. Similarly, the higher ozone abundances correspond to the edge of the polar vortex in both present-day and also future conditions. Hence, we conclude that the "mismatch" between future dynamics and present ozone field mentioned by the reviewer is unlikely to affect the results.

We have included a discussion of this issue in the main text as follows: *We also evaluated potential inconsistencies arising from the use of a fixed (constant) 3-D ozone climatology (which is derived over 2005-2020) in an ensemble with a time-evolving dynamical state (RCP8.5_fO$_3$). The very small projected changes in the vortex morphology in our two CCMs under RCP8.5 ensures spatial coherence between the vortex and the 3-D ozone climatology even in the late stages of the simulations (2080-2099), thus minimizing any potential artifacts.*

**RC 3.2**  2. One way to avoid the above caveat is to consider an interactive "no-recovery" run with ODS fixed at present levels. The comparison of such run with the "recovery" run will demonstrate a true effect of Arctic ozone recovery on the climate. Such a "no-recovery" run will nevertheless include Arctic ozone increase due to intensified BD circulation, making the difference in ozone, and also in ozone induced effects, smaller than what is shown in the present paper. In fact, I am surprised the authors did not consider fixed-ODS simulations which should be available from CCMI data sets for both SOCOL and WACCM. I strongly recommend including such CCMI runs to the paper. If it turns out that the difference between recovery runs and fixed-ODS runs won't be large in the Arctic I would be cautious in using term "influence of ozone recovery" in the title of the paper because ozone recovery implies a recovery from the effect of ozone depleting substances and not necessarily the influence of GHG increases.

[Figure]

Figure R1: **Impact of the interactivity of ozone on the mean state in the two CCMs**. Annual mean differences between runs with interactive (RCP8.5) and prescribed ozone climatology (RCP8.5_fO₃) in temperature (top) and zonal mean zonal wind (bottom) for SOCOL (left) and WACCM (right), averaged over the "early" period of the simulations (2005-2019). Stippled regions are not significant at the 95% level. Units K.

[Figure]

Figure R2: **Coherence between vortex shape and imposed ozone climatology**. Springtime ozone climatology at 50 hPa used for the RCP8.5_fO$_3$ experiment (color shading), with superimposed geopotential height (in units of km) corresponding to the same pressure level (isolines) for WACCM (top) and SOCOL-MPIOM (bottom), averaged over the "early" (2005-2024; left) vs "late" period (2080-2099; right). The lowest isolines (19.5-19.7 km) identify the approximate location of the stratospheric polar vortex, while the meridional gradient in geopotential heigh corresponds to its strength. As one can see, the lowest ozone concentrations that are imposed in the fO$_3$ ensemble correspond to the location of the vortex in both time periods.

**AR 3.2** We thank the reviewer for their useful recommendation. First, we wish to clarify, as pointed out in our replies to Ref#2, that our main focus is on the dynamical impacts of the overall long term ozone changes in the Arctic. Long-term future changes in Arctic ozone will be largely caused by both ODS and GHG emissions; the runs suggested by the referee (the runs that were formerly termed "no-recovery" with ODS fixed at present levels) would only isolate the ozone changes due to ODSs alone (i.e. what is commonly terms as "recovery" in the WMO), as well as the climatic and dynamical changes induced by them. As the reviewer correctly points out, some sensitivity runs aimed at isolating the impact of individual drivers (GHGs and ODS) were designed within CCMI-1 (Morgenstern et al., 2017). In particular, there was one set of experiments called "SEN-C2-fODS/SEN-C2-fODS2000", which, if compared against the reference future scenario (refC2), would isolate the role of ODSs. Unfortunately, while the vast majority of CCMI models performed the "SEN-C2-fODS" experiments using ODS fixed at their 1960 levels, only a small fraction of them performed the same run, using ODS fixed at their 2000 levels. The former set of runs would solely isolate the impact of the ozone "super-recovery", while the latter set comes closer to the goal of this paper. Of the models that performed the "SEN-C2-fODS2000" scenario, only one single CCM performed an ensemble with more than 1 realization: CESM1-WACCM. We note at this point that this model is slightly different from the WACCM version used in this paper, as it contains several upgrades including a modification in the gravity wave parameterization (Garcia et al., 2017). We analyzed the response during March-April in this one model, and found a weakening of the polar vortex of 2-3 m/s in refD2 relative to SEN-C2-fODS2000, which is marginally significant near the core of the polar vortex (Fig. R3). Note that the signal is of the same sign (vortex weakening) as the difference between RCP8.5 vs RCP8.5_fO$_3$ (Fig. 3 in main paper). As noted in our reply to Ref#2, the smaller signal is consistent with the smaller ozone changes (and thus forcing) in these CCMI-1 experiments.

We believe this result further supports our conclusions and therefore, we have added this Figure to the Appendix (Fig. A6), and have added a small discussion near L295, as follows: *We also tested the robustness of the dynamical impacts documented here, by analyzing a set of CCMI-1 runs that isolate the impact of ODS-driven ozone recovery (senC2-fODS2000); these runs were performed by CESM1-WACCM and show a signal that is consistent, indicating weakening of the vortex when ozone recovery is present (Fig.A6). However, the signal is smaller in those experiments, due to the smaller ozone forcing.*

**RC 3.3** 3. I strongly suggest providing quantitative information in the abstract when reporting on the effect of ozone recovery. For example, when you say that ozone induced changes cancel out some of the GHG changes it would be good to know the size of the cancellation.

[Figure]

Figure R3: **Impact of ODS in experiments contributing to CCMI-1.** Springtime (March - April) anomalies in zonal mean zonal wind, driven by ODS emissions, as simulated by CESM1-WACCM, quantified as ensemble mean difference between the reference future scenario of CCMI-1 from this CCM (referred to as "refC2" in Morgenstern et al., 2017) and a future scenario with ODS surface-mixing ratios fixed at their year-2000 values (referred to as "SEN-C2-fODS2000" in Morgenstern et al., 2017). Contour lines show the climatological averages over 2005-2025 in the "refC2" ensemble. Changes that are stippled are not significant at the 95% level. Units m/s.

**AR 3.3** Thank you for this suggestion. The exact size of the cancellation depends on the model being considered and metric. We have now added more quantitative statements, whenever possible, including also the uncertainty range spanned by both CCMs (50-100%) and spelling out the metric for which in the abstract around L14, as follows: *In the stratosphere, Arctic ozone changes cancel out a much larger fraction of the GHG induced signal (up to 50-100%), resulting in no overall change in the projected springtime Northern Annular Mode, and a reduction of the GHG-induced delay of vortex breakdown of around 15 days.*

**RC 3.4** Minor comments:

L19 How about QBO effects on ozone?

**AR 3.4** The QBO effects on ozone are not fully considered, but since the QBO does not change in any of our future scenarios (we map the observed QBO cycles over 1954-2009 into the future in both models), this inconsistency is not expected to affect the results. As a matter of fact, the differences between interactive ozone runs (RCP8.5) and runs with prescribed climatological ozone (RCP8.5_fO$_3$) are negligible in the lower stratosphere even in the QBO domain, for both temperature and zonal mean zonal wind (Fig. R1). We have included a note on this in L127-129.

**RC 3.5** L25-26: I miss mentioning of low temperatures which are the key meteorological factor favoring stratospheric ozone depletion

**AR 3.5** This is noted in the previous sentence as follows: *Conditions within the cold, isolated vortex favour springtime chemical ozone depletion....* We have now also adapted the text to mention the need for low temperatures (i.e. by mentioning *cold local temperatures*) in the same sentence, too.

**RC 3.6** L59: Fusco and Salbi (1999) would be a more relevant reference in this context:

**AR 3.6** We have now added a reference to this paper. Thank you.

**AR 3.7** L159: Was 5 m/s the threshold used by Butler and Domeisen (2021)?

**AR 3.7** Yes, it was used by A. H. Butler and Domeisen, 2021: the wording was unclear and has been corrected. We use 7 m/s while they used 5 m/s. This choice was motivated by the strong vortex (and delayed breakdown) bias in our models; we adapted the text accordingly in the methods section at L174 to clarify this point.

**AR 3.8** L274: I think WACCM has insignificant response in the polar vortex (Fig. 1d)

**AR 3.8** We were referring to the wind changes at slightly lower altitudes, which are significant in WACCM. However, we realize that the focus of this discussion is the vortex defined at 10 hPa, where wind changes are indeed non-significant in WACCM: we have adapted the text accordingly, as follows: *...but no significant response in another (WACCM).*

**AR 3.9** L297: What does "fully offsetting any GHG effect" mean?

**AR 3.9** It means that ozone acts in the opposite direction compared to GHGs, leading to an equal (and opposite) response in the stratospheric NAM (NAM-), while GHGs drive a positive NAM change (NAM+). The combined effect (ozone + GHGs) on the stratospheric NAM is therefore small and uncertain. We realize that this sentence might have been confusing to the reader and have reformulated it as follows: *cancelling any GHG-induced effects on the stratospheric NAM, effectively leading to no robust changes in the NAM.*

**Reviewer 1**

**RC 1.1**    Overall, this is an interesting study that indicates a role of future ozone changes on the circulation response which generally acts in the opposite sense to circulation changes induced by greenhouse gases. I think there is some issue with the wording of "ozone recovery" as used throughout (see Major comment) but I think the results overall are compelling and may be suitable for publication after major revisions.

Major

- I agree with other reviewers that this paper may be improved by changing the wording throughout from the response to "Arctic ozone recovery" to something else, even though the authors have tried to more clearly define what they mean by "ozone recovery" on line 133. Since the end of the century is considered, the ozone changes that have occurred by then are beyond the time where changes can be related solely to recovery and instead will be largely due to GHG changes, which will also be strongly scenario dependent. This would be a relatively easy fix, as it's essentially just changing the wording throughout- for example the "Recovery" experiment could instead be called something like "RCP8.5 ozone changes" or something to that effect. The title could simply be "The influence of springtime Arctic ozone changes on stratospheric and surface climate". I think this will vastly improve the understanding and meaning of the work because readers will get less caught up in how "recovery" is defined and what is meant, and the wording will better reflect what is being done.

**AR 1.1**    We thank the reviewer for this useful comment and recommendation. We changed the title to *The influence of future changes in springtime Arctic ozone on stratospheric and surface climate* and wording throughout the manuscript, replacing any references to "Recovery" with "future Arctic ozone changes" in the high-emission scenario considered here (RCP8.5). This is also reflected in the labeling of the experiments, which are now called RCP8.5 (with interactive chemistry, including long-term ozone changes in the Arctic) and RCP8.5_fO$_3$ (with prescribed ozone climatology, obtained from the early-period (2005-2019) of RCP8.5 for each CCM).

**RC 1.2**    Minor comments:

Line 8- 12: These sentences could be better written/clarified, because the beginning of the sentence is "Under the high-emission scenario examined in this work, Arctic ozone returns to..." which describes what happens to ozone due to both changes in ODS and GHG together. But then the sentence goes "Thereby, it warms the lower stratosphere, ..." and here it is not clear whether you are still talking about the response in the RCP8.5 climate, or the part of the response due to ozone alone. Perhaps it should also be stated more clearly that these changes due to ozone alone are generally opposite in sign to the

changes due to GHGs.

**AR 1.2**  We indeed mean the part of the response due to ozone changes alone. We realize that this sentence in the abstract was not clear enough and reformulated it as follows: *Thereby, the increase in Arctic ozone in this scenario warms the lower Arctic stratosphere, reduces the strength of the polar vortex, advancing its breakdown, and weakens the Brewer-Dobson circulation. The ozone-induced changes in springtime generally oppose the effects of GHGs on the polar vortex.*

**RC 1.3**  Line 50-54: The sentence structure is quite confusing here... is the "including e.g. a delay in the breakdown of the stratospheric polar vortex" referring to ozone depletion or ozone recovery? I would assume it is referring to ozone depletion, but then in the next sentence "These dynamical changes extend to the troposphere, resulting in a negative phase of the SAM..." is no longer referring to ozone depletion (presumably) but ozone recovery. I would rethink how these sentences are structured because as is it is confusing which impacts are related to ozone depletion vs ozone recovery.

**AR 1.3**  The reviewer is correct; this sentence was indeed mixing up effects of ozone depletion (delay in the vortex breakdown) and recovery (negative phase of the SAM). We have now broken down this text into two separate sentences, one reporting the effects of ozone depletion, and one reported the effects of ozone recovery, as follows: *Over the recent past, ozone depletion has led to a delay in the breakdown of the stratospheric polar vortex, and a speed-up of the Brewer Dobson circulation (BDC)(Abalos et al., 2019; Polvani et al., 2019). In the future, Antarctic ozone recovery from declining ODS abundances will slow down the BDC (Polvani et al., 2019) and anticipate the breakdown of the stratospheric polar vortex (Mindlin et al., 2021).*

**RC 1.4**  Line 76-77, line 140: This is related to my major comment 1, but these sentences are an example of why "recovery" is not quite accurate here- because the RCP8.5 ozone changes that occur include both ozone recovery + GHG effects (at least by the end of the century). For example, this sentence could instead be phrased "to better understand the role of springtime Arctic ozone changes in a future climate over the 21st century ..., isolating them from the effects of GHGs alone."

**AR 1.4**  We have adopted the wording suggested by the referee. Thank you!

**RC 1.5**  Line 126, line 134, line 240: would be helpful to keep emphasizing, here and elsewhere, that this will show the impact of long-term ozone changes *for one scenario*; as the ozone response (and its influences on dynamics) may be quite sensitive to which scenario is selected.

**AR 1.5**  We have added a note in all these lines to emphasize that we consider a single (high-emission) scenario. Note that we also emphasize this aspect in the final discussion paragraph in the conclusions section.

**RC 1.6**  Table 1: How much does interactive ozone matter? I imagine a third simulation could be done with prescribed, evolving ozone, and the prescribed and interactive ozone simulations are then compared, which may identify how much ozone-circulation coupling matters vs how much of the response here attributed to ozone recovery is just due to dynamic changes that are also helping to drive ozone changes. Along these same lines, the way that the response to ozone alone is currently derived is assuming (GHG+ozone) – (GHG alone) = ozone alone, but what if there are non-linearities in the response to the first term that may arise from ozone-circulation coupling under climate change?

**AR 1.6**  We thank the reviewer for this insightful comment. The reviewer wonders if the comparison of interactive ozone experiments (which by construction include long-term changes in the Arctic) vs. prescribed ozone experiments (which use a climatological and thus fixed ozone field) truly isolates the impacts of long-term ozone changes, or whether any dynamical changes between the two ensembles, which can in turn modulate ozone recovery (such as a slower or faster BDC), could be "misinterpreted" as an ozone effect. We argue that this distinction would not be important, as long as those dynamic changes that help or counter-act ozone recovery can ultimately be ascribed to ozone. The design of the climatological ozone runs, which includes chemical feedbacks from methane oxidation, ensures that the background state is very close to that of the interactive ozone runs (see e.g. Friedel et al., 2022a for details); thus, comparing both ensembles truly isolates the effects of long-term ozone changes. Moreover, including five members should ensure that we properly separate the effects of ozone from those due to internal variability.

Running another ensemble with time-evolving ozone forcing would indeed complement our experiments. However, they would also come with their own caveats, such as inconsistencies between the prescribed ozone field runs and the circulation raised by Ref#3 (see RC 3.1); in this case, a time-evolving ozone would also contain inter-annual (dynamical) variability, exacerbating such inconsistencies. Also, any feedbacks between ozone and the circulation would not be captured; as an example, the effects of ozone on the BDC (Fig. A5 in the Appendix) and feedbacks on ozone itself cannot be resolved in a simulation prescribing ozone.

Lastly, the reviewer brings up a valid point concerning the importance of non-linearities in the response to combined forcings (GHG + O3) in the RCP8.5 scenario. Unfortunately, it is presently unfeasible for us to add another full set of transient experiments (to have the same statistics, we would need another 5 members for each model!). As an alternative, we tested this effect in a 4-member ensemble with prescribed ozone covering 25 years (thus to-

[Figure]

Figure R4: **Impact of ozone forcing in experiments using constant GHG boundary-conditions.** Springtime (March - April) response in zonal mean temperature (a) and zonal mean zonal wind (b) induced by early vs. late 21st century ozone forcing in the RCP8.5 scenario. These differences are derived from two 100-year time-slice coupled experiments from WACCM with fixed GHGs (2075), imposing two different ozone forcing climatologies: one which is consistent with the underlying GHGs (2075-2089 from the RCP8.5 ensemble) and one which is derived from present-day levels (2005-2019 from the RCP8.5 ensemble). Contour lines in (b) show the climatological averages over 2075-2089 RCP8.5 ensemble. Changes that are stippled are not significant at the 95% level. Units are K and m/s.

taling 100 years) from WACCM, initialized under 2075 conditions and persistent boundary conditions for that year (2075) and a fully coupled ocean. We performed these time-slice experiments with two different ozone climatologies: one with an ozone climatology derived from present-day (2005-2020 average from the corresponding RCP8.5 ensemble), and one with an ozone climatology derived over the last segment of the RCP8.5 ensemble (2080-2099 average from the corresponding RCP8.5 ensemble). The differences in zonal mean zonal wind between both ensembles are shown in Fig. R4. As one can appreciate from this figure, the impacts on stratospheric climate are very similar in magnitude to those reported in the paper, quantified as differences between transient runs with interactive (RCP8.5) vs climatological ozone (RCP8.5_fO$_3$). Hence, we conclude that the interactivity of ozone under these future scenarios is not crucial for modeling future changes in the Arctic stratosphere and thus any non-linearities arising from ozone-circulation coupling under climate change are of second order importance.

**RC 1.7** Line 150: you may need to subtract the global-mean Z10 at each time step has EOF1 may not reflect the NAM in the future but instead start to capture the changing heights due to climate change- see Gerber et al. 2010: `https://agupubs.onlinelibrary.wiley.com/doi/full/10.1029/2009JD013770`

**AR 1.7** We appreciate the reviewer's suggestion. However, we note that the effects of long-term ozone changes on the NAM is precisely the aim of our study, since we are interested in mean state changes of the atmospheric circulation, rather than the variability. We also note that the IPCC-AR6 reports mean changes in the annular modes (NAM/SAM) in Figs. 4.17 and 4.30. There, they use an even simpler definition, involving zonal mean surface pressure differences between 35°N-65°N, without removing any long-term trends in surface pressure which are, for example, due to Arctic amplification.

**RC 1.8** Figure 1 and Line 252: To be clear- is the RCP8.5 response the same as the "Recovery" response, or is there a difference?

**AR 1.8** Indeed, the RCP8.5 response is what we formerly called "recovery" response. Note that we have now changed this labeling throughout the manuscript and refer specifically to the scenario being used in the interactive ozone ensemble including the effects of long-term ozone changes (RCP8.5). In the former "no recovery" case, we were referring to a situation in which ozone is climatological (i.e. "fixed") and is now referred to as RCP8.5_fO$_3$. We hope the new labeling of the experiments and figures will make the figures more self-explanatory.

**RC 1.9** Line 290: Here Fig 1a-b is referenced but doesn't Fig 1 show the "Recovery" run (influence of both GHGs and ozone)? May be worth having an equivalent Fig 1 in the Appendix for the "No recovery" simulation that could be reference instead, since that is what is being discussed here.

**AR 1.9** Indeed, we were referring to an ensemble (formerly referred to as "Recovery" ensemble) which contains both the effects of GHGs and long-term ozone changes. We have followed the reviewer's recommendation, and have now added the equivalent of Fig. 1 in the Appendix (Fig. A2), for the RCP8.5_fO$_3$ ensemble (i.e. the equivalent of what was formerly called "No recovery" scenario).

**RC 1.10** Line 302: According to Butler and Domeisen (2020) and Butler et al (2019), both early and late FSWs induce a shift towards negative NAO (but the timing determines when the negative NAO occurs).

**AR 1.10**  Indeed, it has previously been shown that both early and late FSWs shift the NAM in the direction of its negative phase. However, Thiéblemont et al., 2019 and Friedel et al., 2022b have shown that late FSWs are usually preceded by a positive NAM (at least in the models analysed in these studies), and the FSW in these cases "neutralizes" the surface NAM+ signal rather than inducing a negative phase of the NAM. We have added the suggested references and adapted the sentence accordingly, as follows:

*"as both early and late FSWs are usually associated with a shift of the surface NAM in spring towards its negative phase(Black et al., 2006; Ayarzagüena and Serrano, 2009; Thiéblemont et al., 2019; A. H. Butler et al., 2019; A. H. Butler and Domeisen, 2021)".*

**RC 1.11**  Line 321: It's not clear to me what is being shown in Fig 5 a/d... it says something about the "ozone-induced temperature changes", does that mean this is Recovery-No Recovery and if so why is it labeled RCP8.5? And if so why does it not seem to match Fig 3? The caption could be more clear and the text here could better explain what is being shown.

**AR 1.11**  Indeed, Fig. 5 (panels a and d) show the "total" ozone-induced temperature changes, quantified as simple difference "Recovery" minus "No Recovery" (what we now call RCP8.5 and RCP8.5_fO$_3$, respectively). Note that the mathematical notation in a/b shows the variable being plotted ($\delta$ T), the driver ($\delta$ O$_3$) and where this driver is derived from (RCP8.5). In other words, the RCP8.5 label identifies where the long-term ozone changes are coming from. And these two panels are indeed identical to Fig. 3 a-b: they are just displayed on slightly different color scales.

The other panels of Fig. 5 (panels b/c/e/f) are aimed at displaying the contribution of individual processes (e.g. radiation vs dynamics) to the total temperature changes shown in panels a and d, derived by means of offline PORT calculation. We have rephrased and extended the figure caption to more accurately describe what is being shown.

**RC 1.12**  Line 391: What is meant by "a correct representation of the ozone recovery"? (how can we know which future projection is "correct"?). Along these lines... since many if not most CMIP5 models used prescribed ozone (not interactive)- what implications are there given the results of this study? i.e., those models should all have the exact same influence induced by Arctic Ozone, suggesting that for those models' ozone response would not be an additional source of uncertainty (e.g., differences in the ozone response could not explain the difference in polar vortex response across those models). This may be worth pointing out/discussing.

**AR 1.12**  We thank the reviewer for their comment. First, what we mean with "correct representation" of long-term ozone trends is that models should include an ozone forcing

data-set that is consistent with external forcings, including GHGs. This is now increasingly the case in CMIP6 models, but was not in CMIP5 models (see e.g., Maycock, 2016). This is what we mean with "correct representation" of the long-term ozone trends.

The second point raised by the reviewer concerns the wider implications of this work, and in particular the role of ozone as an additional source of uncertainty. As (correctly) stated by the reviewer, if all models had the **same exact influence** by Arctic ozone (i.e., a shift rowards a negative state of the stratospheric NAM), then indeed the ozone would not be a source of uncertainty; instead, the uncertainty would originate in their response to GHGs and/or other external forcings. However, while the sign of the ozone-induced changes would likely be robust uncertain across models, we think that the magnitude will likely be model dependent due to two reasons. First, there is uncertainty in terms of the recovery rates across models, as one can appreciate from Fig.5 in S. S. Dhomse et al., 2018. CCMs can differ by a factor of two in the projected springtime Arctic changes in 2100, relative to 2000. Hence, ozone may well represent a source of uncertainty at least in CCMs, as CCMs with the largest increase in Arctic ozone abundances may also be the models with the largest influence of ozone on their projected vortex changes. The polar vortex strength changes documented in this paper are in the order of 4 - 6 m/s: this represents a good fraction (40 - 50%) of the CMIP6 inter-model uncertainty in springtime (MAM) shown in Karpechko et al., 2022 (their Fig. 2), and CCMs constitute a good fraction (1/3) of CMIP6 models. Second, models may also have different sensitivities to the ozone forcing, again creating uncertainty. As an example, even models without interactive chemistry imposing the same ozone forcing can exhibit different dynamical responses (Lin et al., 2017). Hence, ozone could create uncertainty even in models without interactive chemistry. Considering all these issues, we would caution against the reviewer's statement that "ozone could not explain the difference in polar vortex response across those models".

Following the referee's suggestion, we have added a few sentences to discuss this point in the conclusions section, as follows: *The implications of this study are threefold. First, models need to include a consistent representation of stratospheric ozone trends; models without interactive chemistry thus need to impose a forcing-consistent ozone data-set Maycock2016. Second, in models with interactive chemistry (CCMs), any uncertainty in the projected recovery rates could result in uncertainty in the degree to which ozone offsets the effects of GHGs in these types of models. Third, models may exhibit different sensitivities to long-term ozone trends (see, e.g. lin2017dependence). Considering these aspects and in particular constraining any of these potential sources of uncertainty might help to decrease the longstanding uncertainty about the dynamical effect of a given radiative forcing....*

**RC 1.13**   Technical

Line 32: mitigating -¿mitigate

Line 33: remove "as"

Line 36: can remove second "stratosphere" after "Antarctic" as stratosphere is already specified

Line 47: "still subject" -¿ still a subject

Line 48: may be clearer to say "the downward influence of ozone recovery on the region with the largest..."

Line 54: More recent references could be included here, e.g. Banerjee et al. 2020

Line 56: specify "warmer climatological stratospheric air temperatures"

Line 58: Charlton and Polvani (2007) reference- is this the most appropriate paper for this sentence?

Line 95: "model" -¿ should this be "model ozone" or just "ozone"?

Line 98-99: change semi-colon to colon, add period after references

Line 108: do you mean, you run two "experiments" as listed in Table 1? (as is it could be interpreted as you run two ensembles of 5 members each for each experiment in Table 1).

Line 115: "as ensemble mean" -¿ "as the ensemble mean"

Line 148: "large-scale changes" -¿ "large-scale dynamical changes"

Line 223: "inducing" -¿ "which induces"

Line 238: was "hODS" mentioned before, and what is the meaning?

Line 276: "We note that signal" -¿ "we note that the signal"

Line 289, 313: replace semi-colon with period

Line 295: "on an ensemble mean" -¿ "in the ensemble mean"

Figure 5 caption: the panels referred to in the first sentence of the caption don't match with the text/what is being shown

Line 323: "panels" is misspelled

Line 330, 370: I don't think "anticipating" is the correct word here

Line 334: two "first"s in this sentence, could remove one

**AR 1.13**   We have included all these changes. Thank you.

**References**

Abalos, M., L. Polvani, N. Calvo, D. Kinnison, F. Ploeger, W. Randel, and S. Solomon (2019). "New insights on the impact of ozone-depleting substances on the Brewer-Dobson circulation". In: *Journal of Geophysical Research: Atmospheres* 124.5, pp. 2435–2451.

Ayarzagüena, B. and E. Serrano (2009). "Monthly characterization of the tropospheric circulation over the Euro-Atlantic Area in Relation with the Timing of Stratospheric Final Warmings". In: *Journal of Climate* 22.23, pp. 6313–6324. DOI: 10.1175/2009JCLI2913.1.

Black, R. X., B. A. McDaniel, and W. A. Robinson (2006). "Stratosphere–troposphere coupling during spring onset". In: *Journal of Climate* 19.19, pp. 4891–4901. DOI: 10.1175/JCLI3907.1.

Butler, A., J. S. Daniel, R. W. Portmann, A. Ravishankara, P. J. Young, D. W. Fahey, and K. H. Rosenlof (2016). "Diverse policy implications for future ozone and surface UV in a changing climate". In: *Environmental Research Letters* 11.6, p. 064017.

Butler, A. H. and D. I. Domeisen (2021). "The wave geometry of final stratospheric warming events". In: *Weather and Climate Dynamics* 2.2, pp. 453–474.

Butler, A. H., A. Charlton-Perez, D. I. Domeisen, I. R. Simpson, and J. Sjoberg (2019). "Predictability of Northern Hemisphere Final Stratospheric Warmings and Their Surface Impacts". In: *Geophysical Research Letters* 46.17-18, pp. 10578–10588. DOI: https://doi.org/10.1029/2019GL083346.

Dhomse, S. S., D. Kinnison, M. P. Chipperfield, R. J. Salawitch, I. Cionni, M. I. Hegglin, N. L. Abraham, H. Akiyoshi, A. T. Archibald, E. M. Bednarz, et al. (2018). "Estimates of ozone return dates from Chemistry-Climate Model Initiative simulations". In: *Atmospheric Chemistry and Physics* 18.11, pp. 8409–8438.

Friedel, M., G. Chiodo, A. Stenke, D. I. Domeisen, S. Fueglistaler, J. G. Anet, and T. Peter (2022a). "Springtime arctic ozone depletion forces northern hemisphere climate anomalies". In: *Nature Geoscience* 15.7, pp. 541–547.

Friedel, M., G. Chiodo, A. Stenke, D. I. Domeisen, and T. Peter (2022b). "Effects of Arctic ozone on the stratospheric spring onset and its surface impact". In: *Atmospheric Chemistry and Physics* 22.21, pp. 13997–14017.

Garcia, R. R., A. K. Smith, D. E. Kinnison, Á. de la Cámara, and D. J. Murphy (2017). "Modification of the gravity wave parameterization in the Whole Atmosphere Community Climate Model: Motivation and results". In: *Journal of the Atmospheric Sciences* 74.1, pp. 275–291.

Karpechko, A. Y., H. Afargan-Gerstman, A. H. Butler, D. I. Domeisen, M. Kretschmer, Z. Lawrence, E. Manzini, M. Sigmond, I. R. Simpson, and Z. Wu (2022). "Northern Hemisphere Stratosphere-Troposphere Circulation Change in CMIP6 Models: 1. Inter-Model Spread and Scenario Sensitivity". In: *Journal of Geophysical Research: Atmospheres* 127.18, e2022JD036992.

Lin, P., D. Paynter, L. Polvani, G. J. Correa, Y. Ming, and V. Ramaswamy (2017). "Dependence of model-simulated response to ozone depletion on stratospheric polar vortex climatology". In: *Geophysical Research Letters* 44.12, pp. 6391–6398.

Maycock, A. (2016). "The contribution of ozone to future stratospheric temperature trends". In: *Geophysical Research Letters* 43.9, pp. 4609–4616.

Mindlin, J., T. G. Shepherd, C. Vera, and M. Osman (2021). "Combined effects of global warming and ozone depletion/recovery on Southern Hemisphere atmospheric circulation and regional precipitation". In: *Geophysical Research Letters* 48.12, e2021GL092568.

Morgenstern, O., M. I. Hegglin, E. Rozanov, F. M. O'Connor, N. L. Abraham, H. Akiyoshi, A. T. Archibald, S. Bekki, N. Butchart, M. P. Chipperfield, M. Deushi, S. S. Dhomse, R. R. Garcia, S. C. Hardiman, L. W. Horowitz, P. Jöckel, B. Josse, D. Kinnison, M. Lin, E. Mancini, M. E. Manyin, M. Marchand, V. Marécal, M. Michou, L. D. Oman, G. Pitari, D. A. Plummer, L. E. Revell, D. Saint-Martin, R. Schofield, A. Stenke, K. Stone, K. Sudo, T. Y. Tanaka, S. Tilmes, Y. Yamashita, K. Yoshida, and G. Zeng (Feb. 2017). "Review of the global models used within phase 1 of the Chemistry–Climate Model Initiative (CCMI)". In: *Geoscientific Model Development* 10.2, pp. 639–671. DOI: 10.5194/gmd-10-639-2017.

Polvani, L. M., L. Wang, M. Abalos, N. Butchart, M. Chipperfield, M. Dameris, M. Deushi, S. Dhomse, P. Jöckel, D. Kinnison, et al. (2019). "Large impacts, past and future, of ozone-depleting substances on Brewer-Dobson circulation trends: A multimodel assessment". In: *Journal of Geophysical Research: Atmospheres* 124.13, pp. 6669–6680.

Rao, J. and C. I. Garfinkel (2021). "Projected changes of stratospheric final warmings in the Northern and Southern Hemispheres by CMIP5/6 models". In: *Climate Dynamics* 56.9-10, pp. 3353–3371.

Thiéblemont, R., B. Ayarzagüena, K. Matthes, S. Bekki, J. Abalichin, and U. Langematz (2019). "Drivers and surface signal of interannual variability of boreal stratospheric final warmings". In: *Journal of Geophysical Research: Atmospheres* 124.10, pp. 5400–5417.

---

## Author Response (AR2)

**Author's final response:** "The influence of future changes in springtime Arctic ozone on stratospheric and surface climate"

Chiodo, Friedel, Seeber et al.

**[RC 1.1]** Ar1.7 response: I had suggested being careful with the calculation of EOF1 in a future climate (currently the EOF loading pattern is calculated over the entire period 2005-2099). The authors argue that the IPCC report uses a simpler definition that involves the zonal mean surface pressure differences between 35N and 65N without removing any trends ahead of time. However, I think my point was missed (partly my fault; I suggested removing global-mean height value for each day but I think the easier solution is to base the EOF loading pattern on the 2005-2020 period rather than the full period as is currently done). Per Gerber et al. 2010, the EOF method applied to geopotential heights will at some point stop picking up the NAM as the dominant mode of variability, and instead pick up the climate change-related height increases everywhere as the dominant mode. Thus, EOF1 stops being the NAM if some care is not taken in its calculation in future climate scenarios. Note that the EOF1 calculated per Gerber et al. 2010 will still reflect long term changes in the NAM (because the data projected onto the loading pattern will still contain the trend). The IPCC report method likely worked because in that case they are using a difference between the regions where the NAM centers of action maximize, so the relative differences still reflect the NAM pattern. I would suggest using the EOF loading pattern calculated for the 2005-2020 period rather than the full period and then this may ensure there are no issues. As mentioned in the Gerber paper "When long data sets are used, however, one must be careful to define the annular mode patterns and indices in such a way that they always reflect internal variability. McLandress and Shepherd [2009] and Morgenstern et al. [2010a] address this concern in long integrations (similar CCMVal-2 REF-B2 simulations as considered here) by computing the NAM relative to shorter 40 year periods at the beginning and/or end of the integration."

**AR 1.1** We have implemented the changes requested by the referee, by using the 2005-2020 period for the EOF loading pattern calculation. The results are almost identical, as one can see from the revised Fig. 4a. We have added the following in the methods section near L75 accordingly: "...spatial loading pattern is then calculated based on the same time period (2005-2020)" and near L77 "Subsequently, geopotential height anomalies for the whole time period (2005-2099) are…".

**RC 1.2** Line 15-16: here, it says "In the stratosphere," but then goes on to refer to the springtime Northern Annular Mode. Are you referring to the stratospheric NAM here, or the surface NAM?

**AR 1.2** We are referring to the stratospheric NAM. We have now explicitly stated this in the abstract.

**RC 1.3** Line 55: could change "ozone depletion trends over the recent past" to just "ozone depletion" to avoid duplicate phrase in the next sentence.

Line 59: suggest "a shift towards the negative phase of the Southern Annular Mode (SAM)" – it could also be made more apparent that this sentence is referring to future ozone recovery.

Line 324: there are two erroneous "are"s here and two "a"s

**AR 1.3** We have adopted all these minor changes. Thank you.